# Characterizing Microbiomes via Sequencing of Marker Loci: Techniques To Improve Throughput, Account for Cross-Contamination, and Reduce Cost

Joshua G. Harrison,[a] Gregory D. Randolph,[a] C. Alex Buerkle[a]

[a]University of Wyoming, Laramie, Wyoming, USA

**ABSTRACT** New approaches to characterizing microbiomes via high-throughput sequencing provide impressive gains in efficiency and cost reduction compared to approaches that were standard just a few years ago. However, the speed of method development has been such that staying abreast of the latest technological advances is challenging. Moreover, shifting laboratory protocols to include new methods can be expensive and time consuming. To facilitate adoption of new techniques, we provide a guide and review of recent advances that are relevant for single-locus sequence-based study of microbiomes—from extraction to library preparation—including a primer regarding the use of liquid-handling automation in small-scale academic settings. Additionally, we describe several amendments to published techniques to improve throughput, track contamination, and reduce cost. Notably, we suggest adding synthetic DNA molecules to each sample during nucleic acid extraction, thus providing a method of documenting incidences of cross-contamination. We also describe a dual-indexing scheme for Illumina sequencers that allows multiplexing of many thousands of samples with minimal PhiX input. Collectively, the techniques that we describe demonstrate that laboratory technology need not impose strict limitations on the scale of molecular microbial ecology studies.

**IMPORTANCE** New methods to characterize microbiomes reduce technology-imposed limitations to study design, but many new approaches have not been widely adopted. Here, we present techniques to increase throughput and reduce contamination alongside a thorough review of current best practices.

**KEYWORDS** microbiome, high throughput, next-generation sequencing, spike in, internal standard, library preparation, PCR, automation, multiplexing, metabarcoding

Microbiomes have been at the forefront of biological discovery over the past few decades, largely because of ongoing improvements to nucleic acid sequencing technology. Indeed, new sequencing tools have facilitated the expansion of the microbial portion of the tree of life (1), led to widespread acknowledgment of the importance of microbial symbionts (2, 3), and spurred the development of industries to harness microbiomes (4, 5). However, for most laboratories, adopting the latest sequencing approaches is challenging because best practices are constantly evolving, and shifting to new techniques is time consuming. Thus, many biologists resort to established protocols that can be costly and low throughput and can limit the inferences made possible by sequence data.

For example, we used Google Scholar to search papers published since 2019 for the two terms "microbiome" and "16S" (the latter is a common barcoding locus for bacteria). To gauge current typical practices, we examined the first 50 papers returned from this query that used sequencing tools to characterize microbiomes. We also took a

Address correspondence to Joshua G. Harrison, joshua.grant.harrison@gmail.com.

Review of current best practices to reduce costs associated with microbiome metabarcoding studies. New methods to reduce and account for contamination and improve throughput are also presented and discussed.

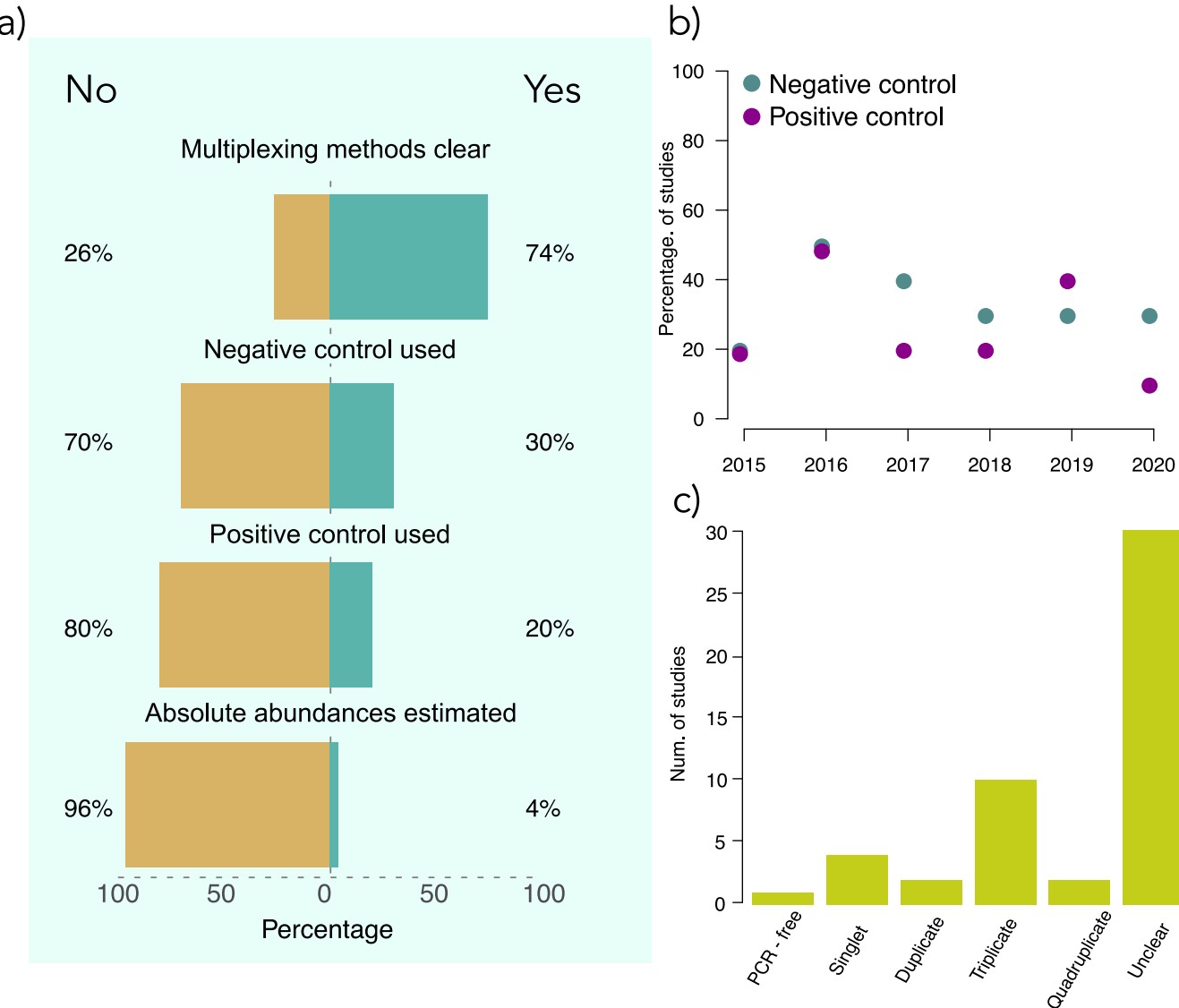

**FIG 1** Snapshot of methodological practices employed from 50 microbiome publications from 2019 to 2020 (see main text). Our goal with this survey was to demonstrate the state of the field. We do not wish to disparage existing methodologies but rather point out that improvements to quality control and throughput are readily possible. (a) Proportion of publications that had clear multiplexing methods and used proper controls. No publication employed an approach that could account for subtle cross-contamination. Additionally, none of the publications reported the use of automation tools, and 41 of the studies used the Illumina MiSeq, which has been superseded by machines with vastly higher output capabilities. (b) Results from a cursory survey of control use in microbiome papers from 2015 to 2020 (see main text for details). (c) PCR replication in the 50 papers surveyed for panel a. More importantly, this panel illustrates that many publications had somewhat unclear methods sections.

cursory look back to 2015 (10 papers per year) to determine if the use of controls has changed over time.

We found that few studies adopted best practices for quality control (Fig. 1). For instance, we found that only 15 of 50 studies used a negative control to account for laboratory reagent contamination (6), and only 10 studies mentioned a positive control of some sort (also see reference 7). There was no obvious trend toward improved inclusion of proper controls with time (Fig. 1b). Fewer studies still, only four, included an internal standard or used quantitative PCR to place compositional relative abundance data from the sequencer on a standard scale to facilitate analysis (see below). Additionally, we found that most studies relied on expensive, but proven, techniques that support relatively limited throughput. For example, Illumina offers several new machines with extreme output (e.g., the NovaSeq), yet 42 of 50 studies used the older MiSeq instrument. Perhaps the most concerning trend we

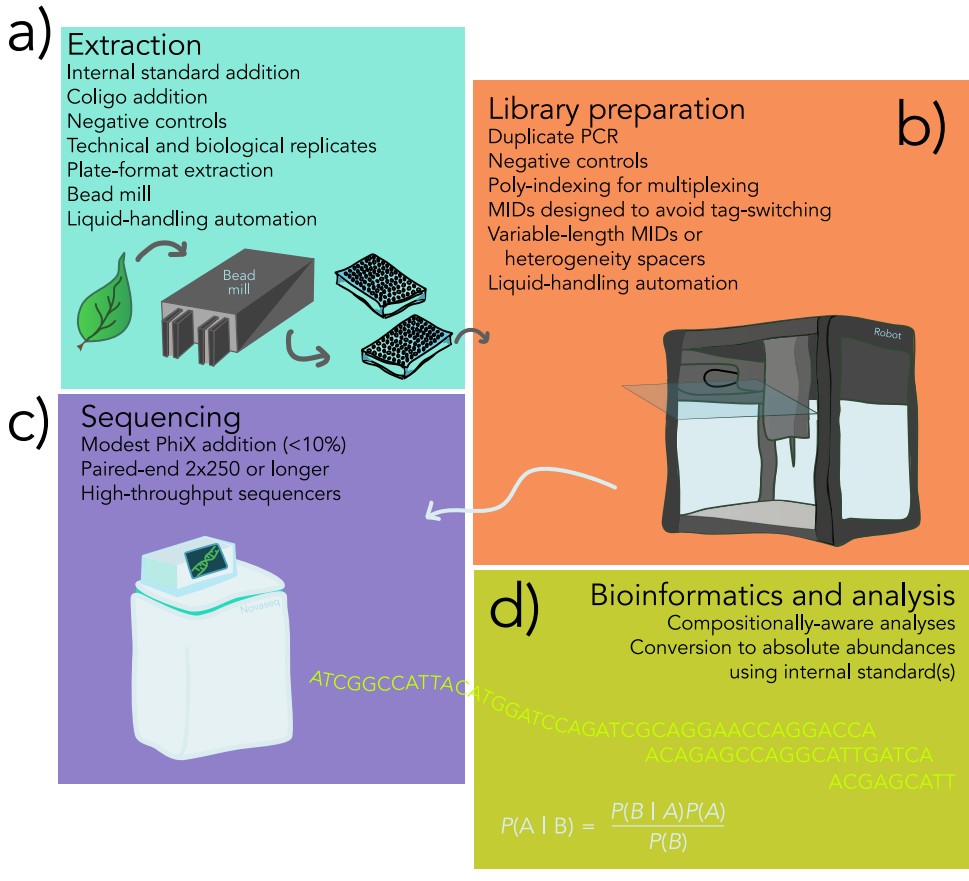

**FIG 2** Overview of best practices to improve throughput and lower the costs associated with amplicon-based characterization of microbiomes. (a) Sample preparation (e.g., weighing tissue) and DNA extraction are time intensive because they are difficult to automate. Bead mills and liquid-handling systems can help improve extraction yield and reduce contamination. Extraction is the ideal time to add internal standards and cross-contamination-checking oligonucleotides (coligos). Due to ubiquitous contamination of extraction reagents with microbes, negative controls are essential. (b) Library preparation is very amenable to automation. New polyindexing strategies, such as the dual-indexing approach we describe here, allow multiplexing of many thousands of samples. Sequencer yield can be improved through ensuring adequate sequence variation at the start of reads. This can be accomplished through the use of variable-length molecular identifiers (MIDs) or heterogeneity spacers. (c) Modern sequencing instruments generate sufficient volume of data for extreme multiplexing, bringing the cost of sequencing down to several US dollars or less per sample. (d) Bioinformatics and analytical techniques are critical to the success of any sequencing project, and the challenge of properly analyzing and storing data should not be overlooked. Users must be aware of the limitations posed by compositional data and curtail inferences as required. Internal standards are of great benefit because they allow relative abundance data to be placed on a scale proportional to absolute abundances.

noticed was that most papers lacked clarity regarding laboratory procedures, thus hampering reproducibility.

While cursory, this survey mirrored our expectations regarding the limitations of typical microbiome laboratory practices that we have observed in reading the current literature. Our goal here is not to disparage the state of the field but rather to draw attention to the opportunity that exists to dramatically reduce costs and improve research outcomes through adoption of new tools and techniques.

Consequently, here, we critically examine every step of single-locus, sequence-based microbiome characterization (often referred to as metabarcoding) (Fig. 2). Our goal is to describe the advantages and disadvantages of new methods while paying specific attention to time and cost-saving techniques (such as automation). Alongside our critical review, we present several improvements to existing protocols for library preparation. When taken together, the techniques we discuss greatly reduce techno-logically imposed limitations on study design. Indeed, we have found that the primary logistical consideration when planning research is no longer the costs associated with

laboratory procedures but instead those associated with sample collection and handling.

Importantly, in this review, we do not discuss experimental design (including sample collection and storage [8, 9]), laboratory inventory management systems (LIMS) (10), or bioinformatic approaches (e.g., see references 11–14). We also do not compare sequencing instruments (including new long-read machines), though the multiplexing advances we describe require the use of the latest generation of short-read sequencers (e.g., the Illumina NovaSeq). While bioinformatics and statistical analysis is beyond the purview of this review, we wish to be explicit that the computational burden incurred by higher-throughput sequencing can be significant and should be considered during the early phases of study design. Notably, as new sequencing platforms are brought to market, existing bioinformatics methods are challenged and can fail; thus, researchers should expect to continually modify their bioinformatic pipeline.

## RESULTS AND DISCUSSION

**Best practices for the characterization of microbiomes—an overview. (i) Robots in the lab: a word regarding liquid-handling automation.** Much of the laboratory work we discuss involves moving small amounts of liquid from one place to another via pipetting. This process is fraught because cross-contamination is a constant threat and variation in pipetting technique among practitioners can influence results. A variety of benchtop robots exist that can automate liquid-handling tasks, including models by Eppendorf, Integra, Opentrons, and others, that cost less than $20,000 USD new (with some simple models costing a fraction of this amount). These instruments consist of a programmable pipette on a movable gantry. Despite their simplicity, they can be extremely useful during nucleic acid extraction and library preparation. More complex robots have large multiposition "decks" that can hold a variety of consumables and additional tools, including, for example, heating or cooling blocks, shakers, centrifuges, or vacuum manifolds (used in place of a centrifuge to pull solutions through columns or filters). These added capabilities come with increased list price—many of these robots are more than $100,000 new.

Notably, the refurbished and used market is large for automation systems. For simple robots, a used machine could suffice, as troubleshooting can be quite straightforward. However, for more complex systems, the benefits of warrantied repair and technical assistance could justify a new purchase, because, in our experience, there will be considerable programming and other technical challenges to surpass. For the motivated and cost-conscious scientist, open-source plans for conversion of three-dimensional (3D) printers to perform liquid handling are available (15, 16) and represent an inexpensive way to experiment with automation (e.g., see reference 17).

When choosing a robot, it is imperative to consider the programming required to accomplish a task. Some machines rely on easy-to-use graphical user interfaces, while others employ proprietary programming languages that are time consuming to learn. Another consideration is error handling, as not all automation systems provide sensible approaches for detecting and reporting errors. Ideally, users will be notified of an error and asked how to proceed. If an instrument does not provide such functionality, its benefits will be undercut, because it will require chaperoning. In the worst case, the instrument will proceed with no documentation of the error and much time will be lost sorting out the mistake. Speaking generally, we have found that robots often require maintenance and troubleshooting, and this should be expected as a probable time cost before purchasing an automation system.

The consumables required by robots are another purchasing consideration. Many liquid-handling systems use proprietary pipette tips (and other plastics) that can add costs. We recommend choosing a robot that can handle both skirted and unskirted 96-well plates as well as 384-well plates. While most protocols used by academic labs rely on 96-well plates, we anticipate a shift to 384-well plates as more researchers seek increased sample throughput (18).

mSystems®

Ideally, robot functionality should allow for a variety of flourishes that can reduce contamination. For example, dispensation speed and height can be reduced to prevent splashing, and pipette tips can be touched to the sides of wells to wick off droplets prior to movement of the pipettor to some other location on the deck. Some robots boast drip detection technology that can warn users of possible contamination. Attention to such details is important, else automation will worsen the threat of cross-contamination.

**(ii) Diagnosing contamination—a ubiquitous problem for microbiome sequencing.** There are two primary types of contamination to consider when performing sequence-based surveys of microbiomes: contamination of samples by foreign microbes and cross-contamination among samples (19, 20). It has become clear that regardless of the care taken by practitioners during laboratory work, contamination is always a threat. This is because microbes are known to occur in many reagents and solvents (6) and are thus unavoidable.

While statistical removal of contaminants has been suggested (e.g., see reference 21) and can be a valuable tool, such approaches should not be substituted for incorporation of negative controls into the design of a study. Moreover, bioinformatic procedures may depend upon data from negative controls (*sensu* the decontam software; Davis et al. [21]). Unfortunately, proper use of negative controls is still surprisingly uncommon (see the introduction above), and the likely prevalence of cross-contamination is particularly concerning (22).

The negative controls required will be determined by study design; however, at minimum, aliquots of all reagents and solvents should be used as template for sequencing (including aliquots from each extraction kit used). Aliquots of reagents should be taken at the end of a laboratory process to maximize the chances of diagnosing contamination that occurred during work. Contamination of negative controls is often tested via PCR; however, we suggest that controls be sequenced, because PCR lacks the sensitivity to characterize instances of minor contamination. Moreover, sequencing allows contaminants to be identified and potentially omitted from downstream analyses.

It is plausible that common laboratory contaminants are present in natural systems; thus, it is potentially inappropriate to remove all taxa that appear in negative controls from a data set. Instead, the abundance of possible contaminants in negative controls versus that in biological samples should be considered. If a taxon occurs with high relative abundance in biological samples but is at low relative abundance in the negative control, then it is likely that the taxon is not solely present due to laboratory contamination. Determining appropriate treatment of contaminants is a topic of ongoing research (19, 21, 23). While bioinformatic and statistical guidance for dealing with contamination is nascent, at a minimum, users can flag possible contaminants and qualify inferences regarding those taxa. For study of low-biomass samples, contamination is a pressing concern (6, 20); however, for those studying systems with high microbial biomass, mild contamination is much less likely to affect inferences.

Most practitioners are now aware of the threat posed by contaminant microbial taxa; however, far less attention has been paid to the specter of cross-contamination. Cross-contamination is potentially more troubling than contamination by nuisance microbes, since the latter type of contamination should occur haphazardly among samples, whereas cross-contamination could be confounded with treatment group (e.g., among samples on a 96-well plate). Therefore, it is important to design laboratory protocols such that cross-contamination can be detected and addressed. While sequencing of negative controls can alert practitioners to catastrophic cross-contamination, such practice does little to indicate the existence of minor bouts of contamination, for instance, when a droplet from a well of a PCR plate migrates to a neighboring well (22).

Tourlousse et al. (24) recently suggested a clever approach for tracking cross-contamination through the use of synthetic oligonucleotides (often referred to colloquially as "oligos"). These authors synthesized 12 unique sequences that were approximately 1,500 nucleotides (nt) long and emulated full-length 16S rRNA genes but that had

negligible similarity to published 16S sequences (for oligonucleotide design, see reference 25). By combining three of these oligonucleotides, 220 unique mixtures can be created. Aliquots of these mixtures can be added to PCR or extraction plates, and the constituent sequences can be used to alert the user to instances of cross-contamination. Tourlousse et al. (24) suggested a way to array mixtures within plates such that neighboring wells are filled with as distinctive mixtures as possible. The downside to this approach is that it only allows approximately 60% of instances of cross-contamination between two samples to be unambiguously detected. For cross-contamination involving three or more samples, detection ability is reduced to ~0.7%.

Accordingly, we have modified the technique described by Tourlousse et al. (24) by designing more and shorter oligonucleotides. We refer to these oligonucleotides as cross-contamination checking oligonucleotides, or "coligos" for short. These coligos consist of a sequence that is complementary to that of forward and reverse primers that bookend a unique sequence taken from the report by Hawkins et al. (26) (for sequences, see Text S1 in the supplemental material). The sequences described by Hawkins et al. (26) allow for detection and correction of insertion, deletion, and substitution events while avoiding extensive internal-complementary-minimizing homopolymers and aiming for reasonably balanced GC content. For most uses, we suggest that 96 coligos is sufficient, as contamination among different 96-well plates is less likely than well-to-well contamination within a plate. However, Hawkins et al. (26) describe $10^{15}$ suitable sequences, if more coligos are desired.

We synthesized 96 coligos that included the popular 515/806 primer pair for 16S and also for the ITS1f/ITS2 primer pair for the internal transcribed spacer (ITS) (192 total). Any primer pair could be substituted for those we chose, including the primers used during genotyping-by-sequencing studies (e.g., see reference 27). Similarly, for transcriptomic studies, coligo sequences could be added to cDNA pools prior to adapter ligation. Coligos can be added at any point during sample processing to good effect; however, we suggest addition prior to DNA extraction (according to reference 24), thus allowing contamination during extraction to be detected. Aside from tracking sample provenance, coligos can also be used to determine if a plate has inadvertently been rotated.

To test their effectiveness, we sequenced a library containing our 96 coligos. Each coligo was added to a single well of a 96-well plate (for additional library preparation details, see Text S1), and three plates were prepared using the two-step PCR approach we describe below. Using the Illumina iSeq instrument, we obtained 2,541,033 reads. We removed the 13 bases immediately following the primer of each forward read (recall that coligos are 13 nt long) and matched these reads to our coligo sequences. We observed negligible contamination among wells—less than 0.01% of all coligo reads were out of place (see Table S1). Cross-contamination, while very minor when it occurred, did occur fairly often; 22% of wells showed some evidence of contamination. When summing read counts across replicates, variation among coligos was approximately normally distributed (see Fig. S2), though substantial variation in read count among technical replicates was observed (Fig. S2) (this variation suggests coligos should not be used as internal standards [ISDs]; see below).

Tourlousse et al. (24) reported variation in read counts among their sample provenance-tracking oligonucleotides as well and suggested that this variation is due to the necessary differences among oligonucleotides in nucleotide composition, including GC content, which have systematic effects on their abundances in sequence reads. This adds to mounting evidence that technically derived variation is a significant source of noise in sequencing data. Indeed, in a recent study of the human gut microbiome, Ji et al. (28) suggested that an abundance threshold exists below which technical variation drives among-sample differences in microbial abundances.

To determine the effectiveness of coligos for typical, likely more complex, libraries from empirical studies, we added them to an Illumina NovaSeq library (2 by 250 created using our two-step protocol; see below) containing over 10,000 replicates

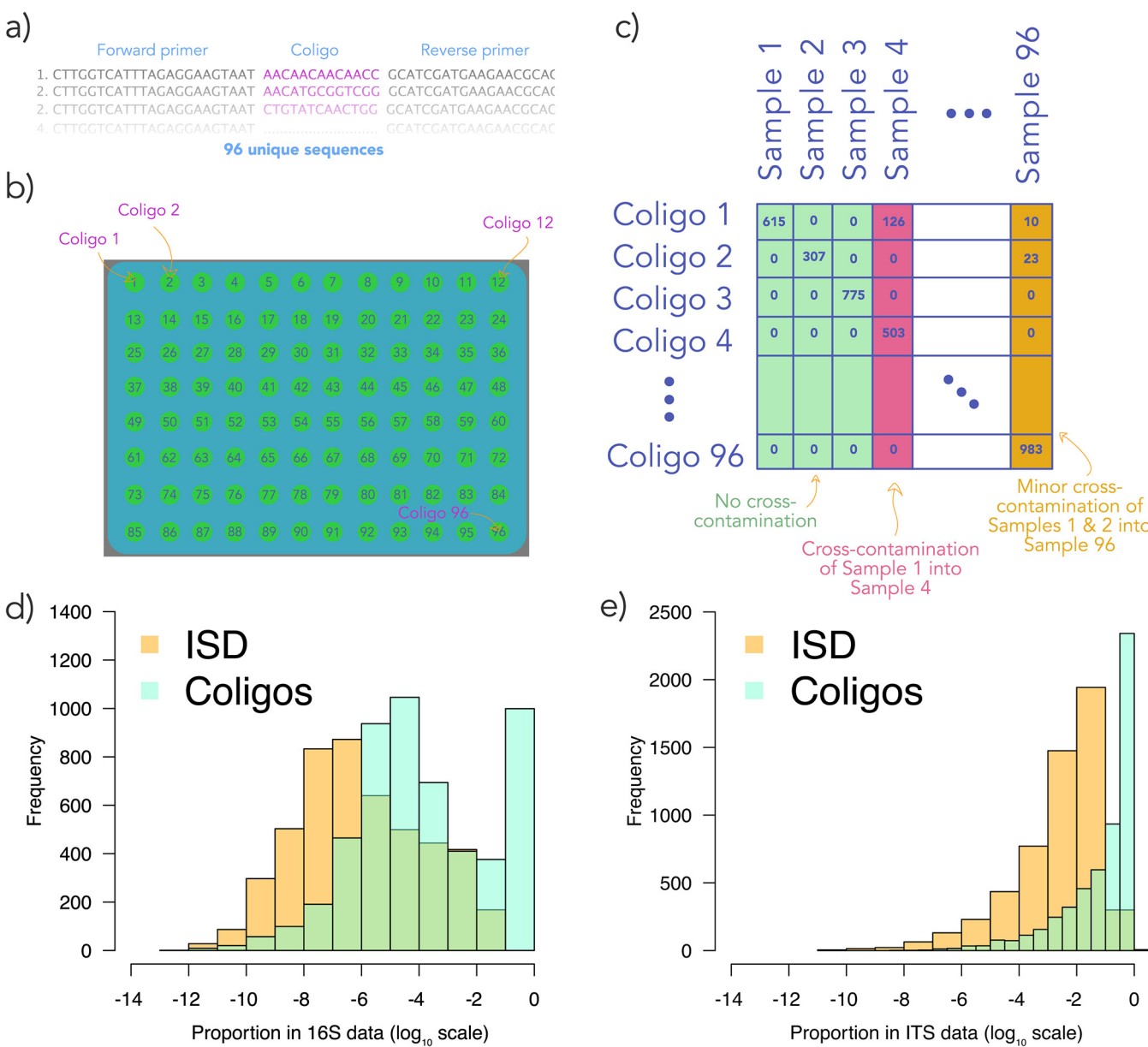

**FIG 3** Cross-contamination occurs when samples are inadvertently mixed during laboratory procedures. We have developed oligonucleotides (which we refer to as coligos) to track incidences of cross-contamination. (a) To make coligos, we placed an identifiable sequence between priming sequences for the marker loci used. (b) Coligos are added to each well of a 96-well plate, preferably prior to DNA extraction. After sequencing, incidence of cross-contamination can be determined. (c) Toy data mimicking an operational taxonomic unit (OTU) table, where samples are columns and OTUs are rows. For these data, samples and coligos are matched such that sample one contains coligo one, sample two contains coligo two, and so on. In this example, samples one through three contain the single expected coligo and thus are uncontaminated. However, sample four contains many reads of coligo one, even though coligo four was the only expected sequence; thus, this sample has been seriously contaminated. Finally, sample 96 has relatively minor contamination by two foreign samples. To demonstrate the use of coligos, we injected them into a complex library composed of both 16S and ITS sequences from a variety of samples (e.g., soil, plant, and water) that differed in DNA concentration and quality. We also included a synthetic DNA internal standard in each library (see main text). The proportion of reads per sample that were associated with coligos and the ISD are shown as frequency histograms; 16S (d) and ITS (e) data are shown separately.

(including molecular identifier [MID]-labeled PCR duplicates from >5,000 samples) collected from various substrates and prepared by different scientists. These samples necessarily varied wildly in the amount and quality of DNA present. We added 0.0016 pg/$\mu$l of coligos to extracted template DNA, prior to normalization of the template to a standard concentration. We observed coligos in ~99.8% of samples. The percentage of reads within each sample ascribed to coligos was generally quite low (on the order of $10^{-8}$% to 1%) (Fig. 3). This result suggests that preferential amplification of coligos, due to their short length, is unlikely to cause undesirable bandwidth capture except,

possibly, for libraries including samples with very little template DNA. In the latter case, coligo and internal standard DNA should be reduced so that the final library is not dominated by nontarget amplicons.

To determine the degree of cross-contamination present, we calculated the median percentage of coligo reads that were present within a sample but that should not have been. Contamination was very minor, with only 0.2% of coligo reads in unexpected wells (see Fig. S3). On the other hand, instances of very minor cross-contamination were common, as 79% of replicates had detectable cross-contamination. We also observed that cross-contamination by multiple samples was common; we observed more than a single foreign coligo in 44% of samples. Contamination was most likely from adjacent wells and decreased with Euclidean distance between wells (see Fig. S4), but contamination from remote wells happened often (as was also found for reference 22). It is plausible that some cross-contamination occurred during synthesis of the coligos.

Given the prevalence of minor cross-contamination that we discovered, we suggest that the use of coligos, or some analogous approach such as that described by Tourlousse et al. (24), become standard for sequence-based studies. While minor cross-contamination will not affect most statistical analyses, given that contaminants are likely only represented by a few reads, it is important to diagnose instances of severe cross-contamination and either remove those samples or otherwise mitigate the influence contaminants could have on inferences. The degree of cross-contamination that is detectable will depend upon the ratio of coligo DNA to template DNA. As more coligos are added, more minor instances of cross-contamination will be detectable. We urge researchers that are interested in using coligos to consider the concentration that we used in our libraries as a starting point and modify that concentration as required.

We note that, ideally, coligos should be added prior to DNA extraction, which we did not do in our simple experiment (because many independent researchers performed extractions using a variety of methods). Determining the appropriate amount of coligo to add to each sample can be challenging, as the amount of DNA in samples, extraction success, and sequencing depth affect the ability to recover reads from coligos. Therefore, the concentration of coligos that we added to each sample could be regarded as a starting point and optimized for a particular study system. If desired, additional coligos could be designed and added to samples during library preparation; thus, contamination during extraction could be distinguished from contamination during library preparation. Finally, we designed our coligos to be very short to reduce monetary cost; however, short sequences do not merge well because of so called "staggering" of reads; thus, we have used forward reads only to determine the incidence of cross-contamination (see the Text S1).

The apparent prevalence of cross-contamination (22) adds further weight to the notion that qualitative analyses (i.e., presence/absence-based analyses and richness) should be undertaken with care when reliant upon metabarcoding data (14, 29).

**(iii) Proper usage of "spike-in" sequences as internal standards.** Data output by current sequencing instruments only provide relative abundance information regarding template molecules. These data are referred to as compositional (30–32) and are a severe limitation of high-throughput sequencing, because biological insight can depend upon accurate measurement of absolute abundances (e.g., see reference 33).

Compositional data arise because sequencers have a finite output; only so many reads are generated per operative period, and those reads are parsed among samples and taxa within a sample. More reads are assigned to taxa that have higher relative abundance. Problematically, when the relative abundance of a taxon increases, it is impossible to know if this was caused by an increase in actual abundance of the focal taxon, a decrease in other taxa, or both (34). This undercuts correlational analyses and, if not corrected, can lead to erroneous inferences (32, 35). Several methods have been suggested to convert relative abundance sequence data into estimates of absolute

microbial abundances (reviewed in reference 34). One such approach is the inclusion of internal standards (ISDs), or "spike-ins," into sequencing efforts.

To use an ISD, a known quantity of a molecule is added to each replicate to be sequenced. The same ISD molecule (or mix of molecules) must be added to each replicate, as any variation within a DNA sequence can affect how well the molecule can be amplified through PCR and sequenced. Since the same amount of ISD has been added to each replicate prior to sequencing, the relative abundances of each taxon ($i$) in a replicate ($x$) can be divided by the relative abundance of the ISD in that replicate using $\frac{x_i}{x_{ISD}}$, thus converting data to a consistent scale. This normalization is effective whether read counts or proportions are used. By placing all taxa on the scale of the ISD, one can determine to what extent a change in the relative abundance of a focal taxon is due to the effect of sampling group (e.g., treatment) rather than a statistical artifact imposed by compositionality. Moreover, multiplication of ratios by the absolute abundance of the ISD (i.e., in cells, moles, or some other unit of abundance) allows the absolute abundance of co-occurring taxa in samples to be estimated (36, 37). A final benefit of an ISD is that it allows explicit statistical modeling of technical variation, which can then be subtracted from among-replicate variation for focal taxa to improve estimates of biological variation (34, 38). The latter benefit is not available when estimating absolute microbial abundance via quantitative PCR (qPCR).

To be effective, an ISD must mirror the behavior of focal microbial taxa during laboratory procedures. This is challenging because every component of a microbial ecology study has the potential to impose taxon-specific bias (e.g., see references 13, 39–42, and 43), and no single ISD can accurately account for these biases for all taxa. Three primary approaches for ISD use exist in the microbiome literature: cellular ISDs, synthetic DNA ISDs, and microbial genomic DNA. Previously, we argued that cellular ISDs or synthetic DNA ISDs should be preferred over microbial genomic DNA (34). Cellular ISDs have the advantage that they could respond to DNA extraction similarly to focal organisms; however, choosing an appropriate culturable organism can be challenging. While synthetic DNA cannot measure variation in extraction success, such ISDs can be highly flexible and cost effective (*sensu* reference 25) and may be more practical than cellular ISDs for many study designs. This is because the synthetic sequence can be optimized to model the focal organism during PCR. Ideally, a simple mixture of ISDs should be included in each sample, with constituents of the mixture designed to mimic focal taxa.

Aside from laboratory biases, various biological and sampling contingencies can undercut the performance of ISDs (reviewed in reference 34), including unaccounted-for differences in sample mass and density, presence of PCR and extraction inhibitors, variation in the lysability of microbial cells (and host cells, if examining endosymbiont assemblages), and copy number variation (Box 1). Thus, careful planning is required to ensure ISDs perform effectively. While adding an ISD at any step prior to DNA normalization provides some benefits, it is best to add ISDs to samples prior to DNA extraction, thus allowing accounting of biases imposed during this step.

As an example of a cost-effective ISD, we shortened one of the synthetic sequences described by Tourlousse et al. (25) from ~1,500 nucleotides to ~170 nucleotides. Short oligonucleotides can be synthesized at lower cost than long oligonucleotides; indeed, for the short ISD we created, enough molar mass for many thousands of samples can be purchased for less than several hundred US dollars. We sequenced aliquots of the ISD that spanned differences in concentration of 5 orders of magnitude and that were mixed with a fixed concentration of DNA from a mock community composed of 10 microbial taxa (for details of library preparation, see Text S1). We observed quantitative behavior of the ISD; that is, as more ISD was added to samples, the proportion of reads assigned to the ISD increased concomitantly (see Fig. S1). We found that two exact sequence variants (ESVs; also referred to as amplicon sequence variants [ASVs]) were associated with the ISD. Tourlousse et al. (25) also reported a proliferation of ESVs for each of their ISDs, depending upon filtering, trimming, and other bioinformatic steps employed. To better determine ISD relative abundance, all ESVs that aligned to

---

**BOX 1: THE PROBLEM WITH COPY NUMBER VARIATION**

The rRNA gene is the standard locus used to characterize microbiomes. Unfortunately, it occurs multiple times throughout the genomes of many microbes and their hosts (44–46), a phenomenon referred to as copy number variation (CNV). CNV is problematic because it means that PCR of the same molar mass of DNA from different organisms will not result in the same amounts of amplicons for each organism. This distorts the relative abundances of sequence counts away from the true relative abundances of the organisms. Several databases (e.g., see reference 47) and software tools exist that provide some insight into CNV for common taxa, in many cases using phylogenetic reconstruction to predict copy number as a character state (e.g., see references 48, 49, and 50). However, copy number can vary both within and among microbial taxa, which undercuts the utility of currently available CNV prediction methods. For instance, Lofgren et al. (46) reported 72 to 156 copies of the ITS among 12 isolates of the fungus *Suillus brevipes*. Since very little is known regarding CNV in the natural world, predictions derived from phylogenetic reconstruction should be regarded as hypotheses and are likely inaccurate. Indeed, Louca et al. (51) demonstrated that most CNV correction tools performed very poorly for the majority of taxa.

Because among-taxa CNV of standard metabarcoding loci is commonplace, it is not generally possible to compare the absolute abundances of different microbial taxa using single-locus sequencing data. However, it is possible to determine shifts in the relative and absolute abundances of a single taxon among sampling groups, assuming that CNV of that taxon is not confounded with the sampling group. For every metabarcoding study, thought should be given to how CNV could be affecting results and biasing inferences.

---

the ISD were summed for each replicate, and this sum was used in proportion calculations. We recommend others use a similar approach.

To demonstrate the use of our ISD in a complex library, we added 0.0005 pg/µl of ISD to each sample within the aforementioned Illumina NovaSeq library (the same library used to test coligo effectiveness). The ISD captured a low proportion of reads (Fig. 3) within each sample but was present in ~91% of samples.

While our approach to ISD design is simplistic and, for many studies, an ISD mixture would be superior, we suggest that the approach described here represents an easy-to-adopt baseline way to circumvent the problems of compositionality within sequencing data (for more, see reference 34).

**Nucleic acid extraction and other preliminaries to library preparation.** Prior to library preparation, nucleic acids must be extracted from cells, PCR inhibitors removed, extraction success quantified in terms of DNA yield, and internal standards added (if desired). Combined, these steps typically require much more time and expense than library preparation and sequencing. Indeed, sample weighing and grinding are often the most time-consuming laboratory steps, as they are difficult to automate (52). Additionally, these steps typically require considerable expenditure of single-use plastic consumables (i.e., pipette tips and microcentrifuge tubes). We are aware of pipette tip-washing tools (e.g., those made by Grenova, Richmond, VA), but these tools are currently unsuitable for pipette tips with filters. Moreover, many pipette tips, because of their filtering inserts, which are typically a different plastic then the body of the tip, are not recyclable, thus contributing to the large amount of laboratory waste generated worldwide (53).

After samples are weighed, DNA extraction can begin. For some sample types, notably plant tissue, mechanical lysis is the first step of the extraction process and, in our

experience, the most critical for obtaining good DNA yield. We suggest that mechanical lysis be considered regardless of substrate (also see references 54 and 55). Grinding can be expedited through the use of a bead mill (e.g., the Qiagen TissueLyser), which is a simple device that shakes tubes containing samples and 2- to 5-mm metal beads (3 mm is our preferred size). Beads can be acid washed and reused. Tungsten carbide beads and stainless-steel beads both work well; however, tungsten carbide beads are more expensive. Steel ball bearings can be purchased extremely cheaply but may not last as long as tungsten carbide beads.

DNA extractions are typically performed either in single-tube or 96-well plate formats; recently, however, various manufacturers have begun offering 384-well plate format kits as well (e.g., the TaKaRa NucleoMag 384 plant kit). Prepackaged DNA extraction kits are available for numerous substrates (e.g., animal and plant tissue, soil, and various culture media). Many of these kits rely on solid-phase extraction (SPE) technology, where nucleic acids are suspended in a high-salt solution and passed through a column, where they are retained within the stationary phase. Once the nucleic acids are bound to the stationary phase, solvents are used to separate and remove unwanted proteins and cellular detritus prior to elution of nucleic acids. This technique can provide high-concentration, pure nucleic acids, but it is time consuming and costly. Notably, these kits can remove PCR inhibitors that are common to certain substrates, such as humic acid in soil and phenols in plants (56). These inhibitors not only undercut PCR but also can reduce the efficacy of internal standards, because they may cause inconsistency among samples in extraction yield (34). A do-it-yourself approach to 96-well plate extraction reliant on Whatman filter paper as a solid phase has been suggested (57, 58). Since a filter paper SPE approach can reduce costs and chemical exposure, it is particularly suited to laboratories interested in substituting monetary cost for time cost and for training laboratories that must minimize exposure to toxins.

Costly SPE kits have been widely used for microbiome studies; however, magnetic bead-based extraction represents an appealing alternative, because it uses fewer consumables and can greatly reduce expense. Briefly, magnetic beads are hybridized to a molecule with affinity for DNA, such as short single-stranded oligonucleotides (59, 60). The beads are then suspended in solution with DNA, which the beads bind to. A magnetic field is used to pull the bound DNA out of solution, allowing contaminants to be washed away. The beads are superparamagnetic, which means they are magnetic only when in the presence of an external magnetic field. A variety of commercially available kits are available. Alternatively, Oberacker et al. (60) provide instructions for the synthesis of magnetic beads consisting of ferrite nanoparticles encased in silica or methacrylic acid. They also provide templates for 3D printed magnet racks that are much less expensive than commercial varieties. Through synthesizing beads, per sample nucleic acid extraction costs can reportedly be reduced to $0.32 (not including plastic consumables).

The bulk of the time involved in DNA extraction, using either SPE or magnetic beads, is spent isolating DNA from PCR-inhibiting compounds. DNA purification has traditionally been required to ensure PCR success; however, improvements have been made to DNA polymerases that allow them to bind to DNA in the presence of inhibitors (e.g., the Thermo-Scientific Phire and Phusion polymerases). So-called "direct PCR" technology uses these improved polymerases to amplify template sequences according to a simple tissue-lysing step (e.g., bead beating or incubation in hot water with degrading enzymes). Direct PCR has been shown to generate similar results to those if traditional extraction techniques (61, 62) but saves a great deal of time and consumables. Recently, Kai et al. (54) reported that mechanical disruption is a necessary preliminary step to direct PCR because of differences in cell wall morphology among microbial taxa, which affect cell lysability and thus extraction yield. Direct PCR requires very little sample mass, which can be a benefit of the approach. Because direct PCR involves little more than a shift to using different polymerases, the adoption of the

technique should not require extensive changes to laboratory protocols and thus deserves more attention.

**(i) Logistical considerations for extraction of microbial DNA.** Each DNA extraction method has its own bias that discriminates against the DNA of certain microbial taxa (e.g., see references 63, 64, and 41). Therefore, to the extent possible, identical extraction methods should be used for all samples and care taken to avoid confounding the extraction technique with the sampling group. Generally, to avoid batch effects, samples should be randomized prior to extraction. We also advocate, if possible, the inclusion of technical replicates during extraction, where the same sample is subdivided and extracted multiple times. Data from these samples can be used to estimate the amount of intrasample biological variation that is present (28).

If possible, positive controls containing cells from taxa of known interest should be subjected to extraction protocols. For exploratory work, a cellular mock community could be included as a positive control. Such mock communities could be made from available culture stocks or purchased (e.g., ZymoBIOMICS provides such an offering). A mock community can also confirm the results of bioinformatics (i.e., the number of ESVs obtained from bioinformatics matches or does not match expectations). Finally, positive controls can be titrated, such that the limit of detection for a particular number of cells or number of molecules can be approximately quantified (65).

DNA extraction is tedious and time consuming. Unfortunately, automation options for DNA extraction that are suitable for most labs are either relatively low throughput, expensive, or require chaperoning (but see reference 66). This is because most extraction techniques require centrifugation to pass solutions through a solid phase or otherwise separate chemical mixtures. The loading and unloading of centrifuges is expensive to automate; consequently, some manufacturers have developed a vacuum manifold system that can pull solutions through a filter. The benefits of such a system include little required oversight after sample loading, assuming the vacuum is strong enough to pull DNA through the solid phase (clogging is a concern when processing soil and other challenging substrates). Alternatively, a simple pipette-on-a-gantry-style system can be used to automate much of the extraction process, with centrifugation steps facilitated by hand. Finally, magnetic bead-based extraction kits may be easier to automate, because magnetic plates are available for many automation systems (or could be custom fabricated).

While we advocate the use of 96-well plates during extraction, we acknowledge that the potential for cross-contamination is high during plate loading. Accordingly, to minimize contamination potential, we suggest suspending dried ground material in lysis buffer prior to transfer to plates via pipetting (this step can be automated). Recently, Custer and Dibner (52) suggested a clever method to avoid contamination during plate loading and eliminate the possibility of double-filling wells through using perforated plate seals and microcentrifuge tubes as funnels to transfer samples.

**(ii) DNA normalization.** It is sometimes desirable to standardize the concentration of extracted DNA prior to PCR and sequencing. Otherwise samples with more DNA are expected to generate more amplicons and more sequence reads. We note that if DNAs are normalized to a standardized concentration, then an ISD should be used if estimates of absolute abundances of microbes are desired (see above). Normalization can be time consuming when the concentration of each replicate is assayed independently (e.g., with a NanoDrop spectrophotometer) and diluted or concentrated manually. Numerous microplate readers are available that can measure each well of a 96-well plate, and some models can measure up to 384 samples at a time. Additionally, multimode microplate readers are available that can measure both absorbance and fluorescence. Fluorescence-based assays are thought to provide more accurate measurements of double-stranded DNA (dsDNA) concentrations than the measurement of absorbance (67), particularly for low-concentration samples. Additionally, spectrophotometric assays are more influenced

by nontarget nucleic acids, such as RNA and single-stranded DNA, than fluorescence methods. This is because fluorescence-based methods rely on dyes with high affinity for dsDNA. A drawback of fluorescence-based assays is that they require additional reagents that add to project costs. When choosing a microplate reader, the minimum volume required for accurate quantification should be considered, as not all devices have the same requirements and repeated quantification can deplete stocks of precious template DNA. To be clear, spectrophotometry and fluorometry measure total nucleic acids present; if measurements of amplicon concentration are desired, then qPCR is a more appropriate approach (68), and none of these tools can accurately estimate cell densities, due to a lack of resolution, CNV (Box 1), and differences in genome size among taxa. Additionally, normalization of samples that contain various amounts of eukaryotic DNA can be challenging, as eukaryotic DNA often will not amplify during PCR.

**Library preparation.** DNA extracted from samples becomes the template from which targeted loci are typically amplified via PCR for detection by sequencing machines. The process of modifying template DNA for sequencing is referred to as "library preparation." Primer choice is a critical consideration when preparing a library and should be in accordance with recommendations for the characterization of focal taxa (Box 2).

---

**BOX 2: CHOOSING PRIMERS FOR PCR**

The majority of single-locus assays of microbial biodiversity rely on sequencing some portion of the rRNA. Often, the V4 region of the 16S locus is chosen to characterize bacteria, and the internal transcribed spacer (ITS) is chosen to characterize fungi. These loci are commonly used because their sequences evolve rapidly enough to distinguish organisms recognized as belonging to different species or variants within species. Sequences of these "barcoding" loci from known organisms are stored in various taxon-specific databases (e.g., SILVA, Greengenes, and UNITE databases [69–71]). Comparison of sequences to these databases allows for detection of known organisms in a sample. Moreover, a taxonomic hypothesis can be generated for a sequence that is not present in these databases by assuming that sequence similarity is predictive of taxonomy. For example, a sequence that is not present in the database but is similar to various known *Actinobacteria* sequences that are in the database can be hypothesized as being from an actinobacter.

While certain primer pairs have become favored among microbial ecologists (e.g., 515/806 for 16S and ITS1f/ITS2 for ITS), every primer pair necessarily imposes some taxonomic bias (40, 72–74). For instance, the first part of the ITS tandem repeat (ITS1) can recover slightly fewer fungal taxa than its counterpart (ITS2), leading Nilsson et al. (13) to suggest ITS2 be adopted as the preferred fungal barcoding locus. Because of the inevitable biases characteristic of a given primer pair, using multiple primer pairs for a study can be beneficial, because it expands the taxonomic breadth that can be surveyed. However, using multiple primers can increase time and consumable costs, and so a balance must be struck between obtaining adequate taxonomic breadth to address the question being asked and study cost.

We also note that the choice of primer and marker locus determines the ideal read length desired from a sequencer. At the time of writing, NovaSeq machines can provide read lengths of up to 250 bases. Even when using paired reads, this may not be sufficient length to recover the whole marker locus from all organisms. In our work, we have noticed that this is a particular problem with using the ITS1 locus for fungi. We often cannot merge paired reads and must resort to concatenating reads or analyzing forward reads only.

Several simple modifications can be made to existing operating procedures to reduce library preparation costs. For example, we advocate performing PCR in duplicate instead of in triplicate reactions (75, 76) and using minimal total reaction volumes. Marotz et al. (75) suggested that singlet PCR is sufficient; however, we have found duplicate PCR to be worth the additional cost, because it provides more assurance that a sample will not be neglected due to minor errors during pipetting. Moreover, if PCR replicates are assigned unique MIDs, then performing PCR in duplicates can provide ample quantification of technical variation induced by library preparation. Typical reaction volumes for PCR range from 20 $\mu$l to 25 $\mu$l. Decreasing the reaction volumes can reduce costs and does not appear to negatively affect results. Indeed, we have achieved satisfactory amplification using a reaction volume of 15 $\mu$l, and Minich et al. (18) have reported good results using as little as a 5-$\mu$l total reaction volume.

When designing library preparation protocols, it can be beneficial to minimize PCR cycles to reduce potential accumulation of technical error due to polymerase infidelity. While the optimal number of PCR cycles will vary with primer choice, template quality, reagents, and so on, Sze and Schloss (77) suggested that cycle count be kept below 35 if possible. High cycle count can also lead to undesirable amplification of very rare template molecules, such as those derived from technical error. Regardless of the methodology chosen, we suggest that greater attention be paid to accurately communicating library preparation methods, as they greatly affect the reproducibility of a study. In our brief survey of literature norms (see the introduction), we often were unable to determine the exact library preparation methods used, particularly when preparation was outsourced to a service provider.

**Multiplexing during library construction.** Typically, researchers wish to sequence many samples simultaneously—a process referred to as multiplexing. Such an approach is made possible by attaching unique, laboratory-synthesized DNA sequences (referred to as oligonucleotides or colloquially as "oligos") to template molecules during library preparation. These oligonucleotides are referred to as molecular "barcodes" (note that this term is also used for amplicons in other contexts, so we prefer the following term) or "molecular identifiers" (MIDs), and because they are sequenced along with template DNA, they allow attribution of sequences to samples (Fig. 4; an additional term for MIDs in the literature is "index"). Here, we discuss several multiplexing approaches and provide a brief overview of their benefits and challenges. We then present a novel method that improves upon existing techniques.

Multiplexing can be accomplished through appending a MID to one end of a template molecule, referred to as "single indexing," or by appending MIDs to both ends of a template molecule. This process is referred to as "dual indexing" and can drastically reduce oligonucleotide costs, because fewer oligonucleotides are required to achieve the desired level of multiplexing (78, 79). For example, two unique MIDs can be arranged at either end of a template molecule in four combinations. Thus, dual indexing allows for $n^2$ combinations of $n$ MIDs. Triple- and even quad-indexing techniques have been described and can allow extreme multiplexing using few MIDs (e.g., see reference 80). Such approaches can be very efficient in terms of oligonucleotide purchase; however, they may require more complex library preparation and bioinformatics. Thus, we suggest that it is simpler to increase the number of dual-indexing MIDs to achieve the desired level of multiplexing.

A primary difference among multiplexing strategies for Illumina instruments is the placement of the MID in the template molecule in relation to the adapter sequence. Adapters are sequences that include regions that bind to the flow cell and thus allow the template molecule to adhere to the flow cell and be sequenced. Illumina adapters also include MIDs and primers that allow the MIDs to be sequenced (see below). More generally, MIDs can be placed within adapters, according to Illumina's design, or between adapters that solely bind to the flow cell and locus-specific primers (Fig. 4).

The Illumina style of dual indexing relies on MIDs that are placed within the adapter sequence so the entire template to be sequenced is of the form (5' end): flow cell binding

# a) One-step PCR

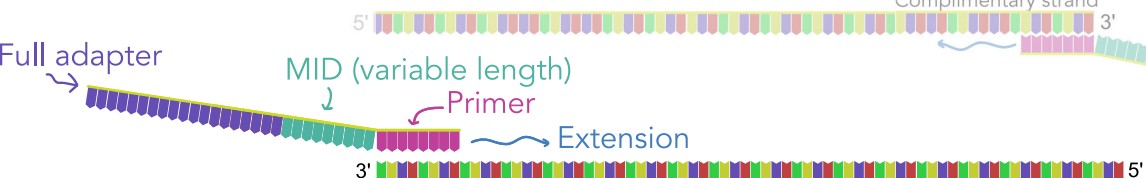

# b) Two-step PCR

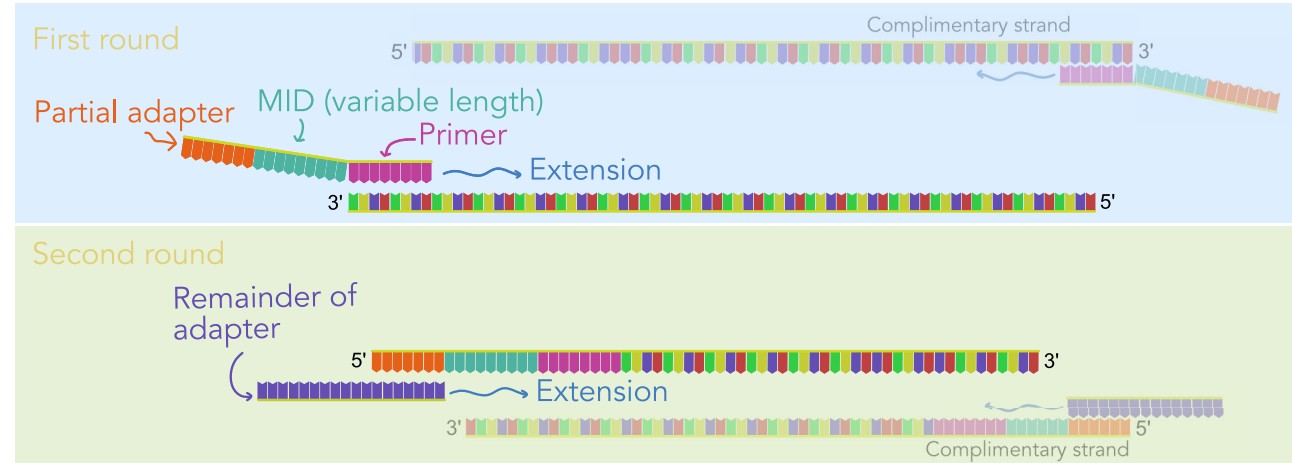

# c) Final amplicon either approach

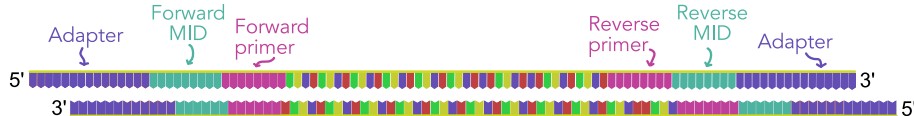

**FIG 4** Visual description of library preparation, which relies on either a one-step (a) or two-step (b) PCR-based procedure. The two techniques differ in the length of oligonucleotides that are required. For most uses, a one-step approach is more cost effective but requires longer oligonucleotides, which only recently became inexpensive and widely available. In the one-step procedure, an Illumina flow cell adapter and molecular identifier (MID; for multiplexing) are added simultaneously when amplifying the template, whereas in the two-step procedure, the template is amplified and a MID sequence is added upstream of the priming region, during an initial round of PCR. A portion of the Illumina flow cell adapter is also added, which serves as the anchor for the second round of PCR, where the remainder of the adapter is added to the amplicon. (c) Both the one- and two-step procedures result in the same amplicon and both rely on dual indexing for multiplexing. We suggest using MIDs that vary in length and that sufficiently differ from one another to allow for unambiguous sample assignment in the event of technically derived sequence variation. Variable-length MIDs inject sequence heterogeneity into the beginning of reads, which improves sequencing performance through prevention of cluster loss on Illumina instruments (see main text).

region–MID (i7)–primer for i7 index–locus-specific primer–template–reverse locus-specific primer–primer for i5 index–MID (i5)–flow cell binding region. During sequencing, the instrument starts at the beginning of the locus-specific primer sequence and performs sequencing by synthesis. Then, during the next "cycle" of the instrument, sequencing begins at the primer for the MID and sequences it; thus, the MIDs are not sequenced simultaneously with the template. This is a general description of Illumina sequencing, and different sequencers and chemistries may vary slightly.

An issue with Illumina's approach is that free adapters within the library can bind to the template along with their associated MIDs, because those free adapters have complementary sequences for the index primers. This can lead to a phenomenon known as "tag switching" or "index hopping," where a MID is assigned to the wrong sample (81–83). This in turn causes sample cross-contamination (also referred to as "cross

talk"). Illumina currently recommends using unique pairings of MIDs at either end of template molecules so that tag switching can be identified and suspect sequences removed from the data (84). While this approach adequately addresses tag switching, it greatly reduces multiplexing capacity.

Alternatively, MIDs can be placed at the 3′ end of the adapter and precede the locus-specific primer (Fig. 4); thus, the MID is internal to the adapter and sequenced simultaneously with the template. Such an approach was devised to improve multiplexing compared to early single-indexing techniques (78, 79, 85). The benefit of this approach is that it is immune to tag switching during sequencing and allows for greater multiplexing with fewer oligonucleotides. The technique's downside is that some portion of the read must be assigned to the MID sequence, because the MID is sequenced along with the template.

Multiplexing techniques also differ as to how they inject sequence heterogeneity into the library. Sequence heterogeneity is beneficial, because a common reason Illumina runs fail to provide the expected output is insufficient variation at the beginning of sequences. If not enough variation is present, the sequencer's detection system cannot distinguish between clusters of molecules being sequenced; thus, those clusters are lost and output is diminished. To prevent this, Illumina recommends a portion of sequencing libraries be composed of random portions of the PhiX bacteriophage genome, which scatters highly variable sequences of balanced purine and pyrimidine content around the flow cell. The recommended amount of PhiX depends upon the complexity of the library to be sequenced and the sequencing instrument, but in some extreme cases, Illumina recommends as high as 50% of the library be composed of PhiX (86). Unfortunately, this means that a concomitant proportion of the sequencer's output will be PhiX sequences that are of no scientific interest; consequently, PhiX addition is a costly way to increase heterogeneity within a library. To reduce the amount of PhiX required, Fadrosh et al. (78) suggested that 0- to 7-nt-long variable sequences ("heterogeneity spacers") be added directly after MIDs, but before locus-specific primers, during oligonucleotide synthesis. Fadrosh et al. (78) report that their approach allowed them to reduce the PhiX component of the library to 10% (also see reference 87).

We suggest an alternative approach that relies on variable-length MIDs to interject sequence variation into libraries. Variable-length MIDs (*sensu* reference 88) obviate heterogeneity spacers, thus reducing the portion of the sequence dedicated to nonbiological data while allowing PhiX input to be reduced. By way of demonstration, we present 96 forward and reverse MIDs for both the 16S and ITS loci (192 unique sequences) (see supplemental material). Our MIDs are a subset of those reported by Gompert et al. (89) and Parada et al. (27) and vary in length from 8 to 10 nt. Sequences were chosen to minimize internal complementarity and homopolymers and allow differentiation using edit distances (all MIDs of the same length are at least a Levenshtein edit distance of two apart from one another [90]). The MIDs are directly followed by sequence that corresponds to the primer region and preceded by the Illumina adapter. We use all 96 by 96 unique combinations of forward and reverse oligonucleotides to support multiplexing of 9,216 samples. If additional multiplexing is desired, then additional MIDs can be designed easily (the extension of MIDs by a few bases leads to many more sequences meeting the aforementioned criteria for distinguishable MIDs [26, 90]]). Incorporating amplicons of multiple loci (e.g., ITS and 16S) can also increase library heterogeneity.

We have incorporated our MIDs into oligonucleotides that also include Illumina flow cell-binding regions and primers for target loci. This allows us to make a library using a single round of PCR, which reduces consumable costs (this is similar to the method used by the Earth Microbiome project [91]). We also present a variant of this protocol that uses shorter oligonucleotides and two rounds of PCR (Fig. 4). The difference between the one-step and two-step protocols is that flow cell adapters are added in a second round of PCR during the two-step approach (Fig. 4). The potential benefit of a two-step protocol is that shorter oligonucleotides can be used than those required by a one-step protocol. Shorter oligonucleotides cost less than longer oligonucleotides, and so it may

be possible to reduce costs somewhat for small batches of samples through the use of a two-step approach (though for large batches of samples, there likely will be no cost savings; see below). Additionally, the primers used to add flow cell adapters during the two-step procedure can be paired with any locus-specific primer. Thus, the two-step procedure is quite flexible and could be useful for research groups that wish to sequence multiple loci or use multiple primers and that wish to avoid the large initial cost of longer oligonucleotides.

We tested our two-step library preparation strategy through sequencing a 16S library containing only the ZymoBIOMICS mock community (Zymo Research, Irvine, CA), which includes eight bacterial taxa and two fungal taxa. We used the Illumina iSeq (paired-end 2 by 150) for sequencing. We recovered all expected taxa from those samples. Subsequently, we sequenced 4,608 samples (each PCR replicate had a unique MID pairing, for a total of 9,216 replicates), consisting of both 16S and ITS amplicons isolated from a variety of substrates (e.g., soil, water, and plant tissue), on the Illumina NovaSeq instrument (this is the same library used to demonstrate coligo and ISD use, as mentioned above). These samples were collected by a number of researchers and extracted using a variety of solid-phase extraction kits. Thus, this library represents the heterogeneity that could be encountered by an active sequencing laboratory. The library was prepared as described in Text S1. We obtained 630,568,959 paired reads that mapped to samples, with an average of 136,842 reads per sample. We obtained more reads for ITS than 16S, but median read count across samples for both loci was high (median read count for 16S, 26,377; median read count for ITS, 123,441). While read count per MID pair necessarily varied depending upon template quality, which varied among substrates and projects, we recovered reads from all samples. We have since obtained similar results from three more libraries of similar complexity and sample count that were sequenced on the NovaSeq instrument.

Given the success of our two-step protocol (which we developed first), we expected the one-step version to provide good results as well. To test this, we sequenced a one-step library containing both 16S and ITS sequences on the Illumina iSeq (paired-end 2 by 150). The library contained DNA extracted from snow, coligos, and ISD. We obtained 3,840,079 reads, which included 800,471 ITS reads and 3,039,608 16S reads (mean of 5,000 sample$^{-1}$ locus$^{-1}$). These results confirm the utility of a one-step PCR approach. By our calculations, adopting a one-step procedure can lead to cost recovery of the initial oligonucleotide expenditure after just a few library preparations (assuming multiplexing of several thousand samples per library).

**Library clean up.** Prior to sequencing, libraries will typically require some form of "cleanup," where unused primers, deoxynucleoside triphosphates (dNTPs), and adapter sequences are removed. Cleanup is particularly important when using Illumina-style dual indexing (see above) to minimize tag switching. Additionally, cleanup can be an effective way to maximize PCR yield when performing two-step PCR, because it prevents unspent primers from the first step of PCR from being amplified during the second round. Unfortunately, library cleanup can be quite costly; so, if the protocol allows, cleanup should be performed only once on pooled MID-labeled DNAs.

Cleanup can be accomplished through chromatography or ethanol precipitation (92, 93), but these methods are cumbersome and time consuming and have fallen out of favor due to their low throughput. A modernized analogous approach relies on the BluePippin instrument (Sage Science, Beverley, MA), which uses a combination of pulsed-field electrophoresis and spectrophotometry to automatically separate and output sequences of a certain size range. Automation of size selection in this way can more precisely select a desired sequence length, is less prone to contamination, and saves time compared to manual gel electrophoresis (94), but it does require specialized equipment and consumables.

Magnetic beads can also be used to perform PCR cleanup (e.g., the popular Axygen kits; Corning, Corning, NY, USA). These beads are nanoscale magnetic particles that bind to DNA. Size selection is achieved via the ratio of beads to DNA; larger DNA molecules preferentially bind to the beads, and as more beads are added, smaller DNA molecules are also bound.

Application of a magnetic field allows the bound DNA to be pulled out of solution. Magnetic bead cleanups do not require specialized equipment (a strong magnetic strip is all that is required); however, the beads are costly. As with bead-based nucleic acid extraction, beads can be synthesized oneself for substantial cost savings. For instance, Oberacker et al. (60) reported that synthesis of beads can reduce costs to approximately $0.05 USD per sample.

Enzymatic PCR cleanup represents an alternative to magnetic bead-based protocols. Cleanup is accomplished through enzymatic degradation of single-stranded DNA and dephosphorylation of surplus dNTPs (95) through the combined action of exonuclease I and shrimp alkaline phosphatase. These enzymes can be purchased pure, and several manufacturers package them as kits (e.g., ExoSAP-IT [Affymetrix, Santa Clara, CA]). When performing cleanup using these enzymes, no loss of template is reported for amplicons of various sizes, including short amplicons approximately 100 nucleotides long (96).

**Sequencing symbionts: how to deal with unwanted host DNA.** The amplification and sequencing of nontarget DNA is a particular challenge for microbial ecologists studying host-associated symbionts. For instance, botanists interested in the bacteria within plants must contend with chloroplast DNA (cpDNA), which is amplified by many commonly used rRNA primers. Indeed, in some cases, 90% or more of reads recovered from plant tissues are cpDNA (97–99). Similar challenges face researchers interested in symbionts within animals, given the abundance of host nuclear rRNA and mitochondrial DNA (mtDNA). Accordingly, researchers have explored three main avenues to reduce nontarget DNA during microbiome sequencing: the use of more selective primers, targeted removal of unwanted sequences from extracted nucleic acid pools, and separation of nontarget cells prior to extraction. For many use cases, such as the characterization of certain taxa, selective primers may be the ideal tool, as they are inexpensive and easily obtainable. However, for exploratory research seeking to broadly characterize microbial assemblages, the taxonomic biases imposed by restrictive primers are undesirable.

Perhaps the most widely used technique by microbial ecologists to avoid amplification of nontarget sequences is peptide nucleic acid (PNA) clamping (100–105). PNAs are oligonucleotides that are complementary to the sequence to be suppressed. The bases in a PNA are attached to a neutral backbone of $N$-(2-aminoethyl)-glycine instead of charged phosphate groups (106, 107). This neutrality allows for a stronger bond between PNAs and single-stranded DNA than what would be experienced during DNA-to-DNA bonding. This allows the link between PNA and its target to persist through PCR, thus blocking the action of DNA polymerase.

Lundberg et al. (101) demonstrated the use of PNAs to suppress plastid DNA in a study of the microbiome of *Arabidopsis thaliana*. Fitzpatrick et al. (108) explored the limits of this approach through sequencing of 32 plant taxa from across the angiosperm phylogeny. In this study, PNA addition was reported to suppress cpDNA for each host taxon, in some cases by up to 65%. However, even single-base mismatches between target sequences and the PNA caused a reduction in performance. A concern with PNA use is that they may impede amplification of target taxa, thus imposing taxonomic biases on sequencing results (e.g., see reference 98).

PNAs can be made to mimic any sequence but are typically used to block either cpDNA or mtDNA. In our own work on plant microbiomes, we have noticed that when PNAs are added to block cpDNA, this can lead to a higher proportion of remaining reads being allocated to mtDNA (unpublished data). Thus, we suggest that researchers consider the use of multiple PNAs to suppress both cpDNA and mtDNA simultaneously.

Several additional methods to selectively reduce the relative abundance of nontarget reads have been proposed but have gained less traction than PNAs among microbial ecologists. For instance, Green and Minz (109) suggest restriction endonuclease digestion of double-stranded DNA created using primers specific to nontarget sequences; the remaining DNA can then be amplified using more general primers. Similarly, Dolinšek et al. (110) use enzymatic oligonucleotides to selectively degrade target RNA. Magnetic bead pulldown methods have also proven effective at reducing nontarget sequences. For

example, Feehery et al. (111) used magnetic beads with methyl-binding domains that specifically bind to a portion of mammalian and fish DNA to reduce the presence of these nontarget molecules (also see reference 112) (commercially available kits include the NEBNext microbiome DNA enrichment kit). Yigit et al. (113) extended this approach to plant tissues and selectively shifted the ratio of nuclear to organellar DNA obtained from five model angiosperm taxa (also see reference 114).

A very different approach to minimizing the concentration of nontarget DNA in libraries is to separate or degrade nontarget cells prior to DNA extraction. For example, the MolYsis kit (Molzym, Bremen, Germany) uses chaotropic solutions to selectively degrade host cells that have weaker cell walls than many microbes (this approach will not work for plants, given the rigid cell walls present in plant tissue). Thoendel et al. (115) suggested that the MolYsis kit outperformed the NEBNext microbiome DNA enrichment kit for removal of nontarget DNA taken from infections of prosthetic joints (also see references 116, 117, and 118). A similar but much more cost-effective approach was demonstrated by Marotz et al. (119) that relies on selective lysis of mammalian cells followed by propidium monoazide treatment (this publication suggests a $0.15 USD cost per sample for this method).

Alternatively, flow cytometry and microfluidics can be used to remove nontarget cells from samples. For example, Wu et al. (120) separated bacteria from human blood cells via microfluidics. Flow cytometry is often used to sort cells by phenotype (121) and can be used to distinguish cells in terms of DNA content (122–124). Such an approach could plausibly be extended to separate eukaryotic cells from microbial cells based on DNA content; however, we are unaware of any studies using this technique. While labor intensive, centrifugation can also be used to separate cells; for example, Utturkar et al. (125) used a combination of centrifugation and cytometry to separate microbial cells from plant cells and allow single-cell genomics of the microbial fraction.

Finally, methods employed by functional genomicists to normalize cDNA libraries could be useful for microbial ecologists. In this context, normalizing refers to manipulating the relative abundances of molecules in libraries to reduce the variation in those abundances. A variety of strategies have been proposed that rely on the activity of different enzymes to selectively degrade the most abundant nucleic acids within a solution (briefly reviewed in reference 126). For example, Zhulidov et al. (127) use a duplex-specific nuclease isolated from Kamchatka crabs that attacks double-stranded DNA. Since complementary sequences of abundant single-stranded DNAs (ssDNAs) are more likely to encounter one another during reassociation, the enzyme can be used to attack these duplexes during a short incubation period and then be deactivated. The resulting library has a higher proportion of low-frequency sequences. The same rationale is applied by Ramond et al. (126) to suppress amplicons of abundant microbial taxa through the activity of S1 nuclease. To our knowledge, the benefits of normalizing DNA amplicon libraries have not been studied thoroughly by microbial ecologists. We suggest these techniques have strong potential to improve qualitative studies of the rare biosphere (128), such as when assaying the presence of rare pathogens (e.g., in wastewater) or dormant taxa.

We note that any method to reduce nontarget sequences will likely impose undesirable taxonomic biases on the resulting library. Consequently, researchers should ensure that suitably complex mock communities are included in libraries to quantify and document these biases.

**Conclusion.** Here, we have provided a methodological "state-of-the-field" assessment for single-locus sequence-based characterization of microbiomes and presented several improvements to existing procedures. Of the techniques discussed, we particularly advocate the rapid adoption of internal standards and methods to account for cross-contamination (such as the "coligo" approach we present here). These technologies are simple and inexpensive to incorporate but can drastically improve experimental outcomes. We also found significant cost and time savings can be obtained through switching to a one-step PCR using our variable-length MIDs.

We acknowledge the challenge of staying up to date with ever-changing sequencing technology. Methods are published weekly, and determining best practices is difficult, particularly for research groups new to sequence-based characterization of microbiomes. Our goal here is to build awareness for methodological advances in hopes that adoption of these techniques will cut costs and improve research outcomes. Better methods can allow for better science, but we urge practitioners to carefully consider the needs of their study. All the techniques we describe here have both advantages and drawbacks, including the not-insignificant time cost of learning and deploying a new protocol. Thus, the best practice of all is to critically appraise a method and determine its suitability for a particular experimental design given logistical constraints.

## MATERIALS AND METHODS

**Data availability.** Oligonucleotide sequences for coligos, a custom demultiplexing script, and the library preparation procedures we discuss can be found at https://github.com/JHarrisonEcoEvo/Genome _Technologies_lab_of_Univ_Wyoming.git.

We provide example sequencing data created using our one-step PCR library preparation procedure at https://mountainscholar.org/handle/20.500.11919/7186. These data were generated by the Illumina iSeq instrument and contain 16S and ITS sequences from snow collected by Abigail Hoffman. At this same URL, we also have provided iSeq data of a library containing only coligos, mock community, and internal standard (at various concentrations). This library was created using our two-step PCR procedure.

Example NovaSeq data generated using our two-step library preparation procedure can be found at https://hdl.handle.net/20.500.11919/7166.

## SUPPLEMENTAL MATERIAL

Supplemental material is available online only.

**TEXT S1**, PDF file, 0.1 MB.

**TEXT S2**, PDF file, 0.1 MB.

**FIG S1**, PDF file, 0.1 MB.

**FIG S2**, PDF file, 0.1 MB.

**FIG S3**, PDF file, 0.1 MB.

**FIG S4**, PDF file, 0.1 MB.

**TABLE S1**, PDF file, 0.1 MB.

## ACKNOWLEDGMENTS

We thank the many scientists whose work we referenced and built upon and two anonymous reviewers for comments on an earlier draft of the manuscript. We also would like to thank the members of the Microbial Ecology Collaborative at the University of Wyoming for letting us use their samples and sequencing data. Computing was performed using the Teton Computing Environment at the Advanced Research Computing Center, University of Wyoming, Laramie (https://doi.org/10.15786/M2FY47). Thanks to Shannon Harris and Muhammad Saqib for assistance with portions of the laboratory work.

This research was supported by National Science Foundation award EPS-1655726.

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
