## [Reviewer comments · mSystems]

Characterizing microbiomes via sequencing of marker loci: techniques to improve throughput, account for cross-contamination, and reduce cost

Joshua Harrison, Gregory Randolph, and C. Buerkle

Corresponding Author(s): Joshua Harrison, University of Wyoming

Review Timeline:

Submission Date:	March 10, 2021
Editorial Decision:	April 12, 2021
Revision Received:	May 26, 2021
Accepted:	June 7, 2021

Editor: Peter Turnbaugh

Reviewer(s): Disclosure of reviewer identity is with reference to reviewer comments included in decision letter(s). The following individuals involved in review of your submission have agreed to reveal their identity: Dieter M Tourlousse (Reviewer #1)

Transaction Report:

DOI: <https://doi.org/10.1128/mSystems.00294-21>

April 12, 2021

Dr. Joshua G Harrison
University of Wyoming
Laramie

Re: mSystems00294-21 (Characterizing microbiomes via sequencing of marker loci: techniques to improve throughput, account for cross-contamination, and reduce cost)

Dear Dr. Joshua G Harrison:

Thank you for your submission, which is a valuable guide for researchers in the microbiome field.

Below you will find the comments of the reviewers.

To submit your modified manuscript, log onto the eJP submission site at <https://msystems.msubmit.net/cgi-bin/main.plex>. If you cannot remember your password, click the "Can't remember your password?" link and follow the instructions on the screen. Go to Author Tasks and click the appropriate manuscript title to begin the resubmission process. The information that you entered when you first submitted the paper will be displayed. Please update the information as necessary. Provide (1) point-by-point responses to the issues raised by the reviewers as file type "Response to Reviewers," not in your cover letter, and (2) a PDF file that indicates the changes from the original submission (by highlighting or underlining the changes) as file type "Marked Up Manuscript - For Review Only."

Due to the SARS-CoV-2 pandemic, our typical 60 day deadline for revisions will not be applied. I hope that you will be able to submit a revised manuscript soon, but want to reassure you that the journal will be flexible in terms of timing, particularly if experimental revisions are needed. When you are ready to resubmit, please know that our staff and Editors are working remotely and handling submissions without delay. If you do not wish to modify the manuscript and prefer to submit it to another journal, please notify me of your decision immediately so that the manuscript may be formally withdrawn from consideration by mSystems.

Sincerely,

Peter Turnbaugh

Editor, mSystems

Journals Department
Reviewer comments:

Reviewer #1 (Comments for the Author):

The authors provide a discussion of the current state-of-the-art in establishing. In addition, new results are presented, focusing on the application of synthetic DNA standards for tracking of sample cross-contamination and "absolute" quantification of. The latter is an increasingly, albeit not yet sufficiently addressed, concern that relates to the compositionality of sequencing data, in the sense that they only provide information about ratios of taxon abundances (or relative abundances). This can be addressed by using the standards as anchors for read count normalization/transformation, such that comparative analyses between samples becomes valid.

Overall, the manuscript is very well written and provides an enjoyable and informative read. Although some parts cover relatively basic topics, I believe that it should be of interest to the reader. I do not have any major concerns/comments but would like to ask the authors to address my questions/comments below. All comments/questions can be considered as minor and should be easily addressable by the authors.

- With respect to the application of coligos for sample tracking and/or cross-contamination detection, it is stated that less than 0.01% of coligo reads were out of place (Line 192). It would be informative to also plot (probably as a Supplementary Figure) the distribution of the percentages of unexpected coligo reads per sample/well. Also, was any consistent contamination observed, this may point to maybe contamination due preparation of the coligo stocks? Further, it is stated that read counts varied across replicates (line 194-196). It is not clear to me how to interpret this; as all wells contain a single coligo (except for cross-contamination), the differences in read counts thus appear to be due to varying read depths per sample/well, rather than variable detection rates (e.g., due to differential PCR amplification) among oligos. In the Tourlousse et al. 2018 paper (Line 197-200), differences between coligos were based on mixtures of three coligos each, such that the observed differences, ignoring compositionality, were attributed to differences in detection efficiencies between coligos in each of the mixtures. Please clarify this. This may also need to be clarified in Figure S2. Also, Figure S2 shows sum coligos with very low count, please briefly discuss; this is again due to differences in library normalization?

- The coligos developed by the authors for sample/cross-contamination tracking are very short. Please discuss its potential for preferential PCR amplification. Although this should not affect the performance for their intended purpose, that may be a question raised by researchers planning to adopt this system.

- At Line 186-187. Please describe the library preparation scheme (two-step/one-step, dual/single indexing). Although this is described in the Supplementary Methods, it helps to readers to get that information without needing to look at the Supplementary Methods.
- At Line 190. It is mentioned that reads are "mapped". Please briefly describe. Btw, it seems that exact matching was used for the short coligos, rather than "mapping".
- Also, it would be good to acknowledge that the level of cross-contamination that can be detected depends on the amount of coligos added to each sample/well, as touched upon in Tournalousse et al. 2018. This seems relevant to the discussion at Line 230-231.
- In line 202, the meaning of 10,000 "replicates" of samples is not clear. You mean 10,000 different samples. Also, please add the type of NovaSeq platform and read length, not critical but just for completeness, as well as the library preparation scheme (two-step/one-step, dual/single indexing).
- In line 208, it seems that proportions (that is, total-sum-scaling to 1) are presented, rather than percentages used in other places (e.g., 0.01% as discussed in the previous comment). Please use to consistent total-sum-scaling basis, preferably as percentages as this may be easier to interpret.
- In Line 306, it is stated that spurious ESVs may be due to degenerate primers. This is not clear to me. The primer sequences should be trimmed prior to analysis and thus not contribute the number of observed ESVs. Please clarify.
- In Line 300-301, it is mentioned that the reads for the IDS showed quantitative behavior, shown as a linear relationship between ISD amount and read counts in Fig. S1. Please explain how the data counts per sample (I guess each different ISD concentration tested represents a different sample/library) were normalized, rarefaction? Also, to my understanding, the relationship between added ISD amount added into a constant amount of mock DNA should display a hyperbolic dependency, from 0% of reads assigned to the IDS to 100% of reads assigned to the IDS. At lower amounts of ISDs added, this will appear as linear. On the other hand, expressing data as the ratio of ISD reads to mock reads should be linear, irrespectively of the range of ISD amounts tested. Basically,

$$\text{isdReads} = \text{isdCopies} / \text{totalCopies} * \text{totalReads}$$

$$= (\text{isdCopies} / (\text{isdCopies} + \text{mockCopies})) * \text{totalReads} \text{ (hyperbole)}$$

$$\text{isdReads} / \text{mockReads} = \text{isdCopies} / \text{mockCopies} \text{ (linear, assuming constant mockCopies)}$$

This is a comment; please consider.

- Line section 448-451, it is suggested that the number of PCR cycles should low, preferably less than 35. While this is true, the actual value very much depends on the polymerase and purity of the sample. Further, in my view, we should talk about reducing the amplicon yield (or fold amplification), rather than PCR cycles. Just a comment to consider.
- Although the use of long reads for analyzing microbiome by full-length rRNA operon sequencing is still in its infancy and may not provide the required throughput, a short paragraph on this topic may be helpful.
- The others often employ NovaSeq for sequencing, because of its high throughput. However, as far

as I know, such sequencing data may be more challenging to handle using state-of-the-art denoisers such as DADA2 as quality scores are binned. The authors should briefly discuss this.

- The authors suggest a read length of 2x250 bp in figure 1; however, not all NovaSeq platforms provide such read lengths. Also, even 2x250 bp reads may be too short for V3V4 amplicons, if read merging is part of the bioinformatics workflow. Please briefly discuss.

- In the Supplementary Material on Bioinformatics, it is stated that reads with more than a single error were removed. I guess this is based on the Expected Error that can be used as filtering criterion, please clarify and mention the term Expected Errors if appropriate as this is quite common.

Reviewer #2 (Comments for the Author):

"Characterizing microbiomes via sequencing of marker loci: techniques to improve throughput, account for cross-contamination, and reduce cost"

This is a review/opinion paper highlighting the challenges of consistency and data reproducibility across microbiome studies. This is an important subject and I think overall the paper does a good job reviewing the current literature on the subject. The paper also highlights the use of a new strategy for spike ins. There are a few parts of the paper which need revised to include additionally important research articles. One section which describes the need for DNA normalization prior to PCR needs to be either eliminated or significantly revised to address shortfalls of this approach. The authors have not made their data publicly available via the standard sequence databases (ncbi or ENA) and thus need to do so before acceptance. Lastly, the authors should consider making their novel oligos and overall primers even more clear and easy for other researchers to simply 'order' and use from IDT. At the current state, it's a little unclear how one would go about doing this as an outsider. I hope they can use this feedback to make the current pipeline more approachable for either new or seasoned microbiome researchers.

Main text:

Abstract Line 11-12:

Interesting that they say nucleic acid extraction or library prep. It seems reasonable that including synthetic molecules at each step is better than one? Could the authors please elaborate? Most well-to-well contamination occurs during DNA extraction (Minich 2019).

Intro

Line 29-42: This is a great paragraph outlining the problems in the field. It would be fantastic if the authors provided a figure showing the results of this query. I understand they have written the detail in the text but a figure would be very appealing to a broad audience. In addition, it would be great if the authors included additional studies in this review dating back over the past 5 year (perhaps an additional 10-20 per year for 5 years) to see if there have been any trends of increased use of controls etc over that timeframe.

Line 39: The newer sequencing instruments (Hiseq 4000, Novaseq, etc) use a patterned flow cell

which does not work as well for low-diversity libraries. With Illumina taking the HiSeq 2500 offline, many researchers still continue to use the miseq as a standard instrument. It might be relevant to point this out as a challenge of doing any amplicon based approach on the Illumina platforms going forward.

Line 73: It would be great if the authors added a table outlining the various robots used in microbiome processing including DNA extraction and library prep along with prices. They do list some manufacturers but don't discuss this in more detail including MSRP list prices.

Line 113-114: There are a few good reviews out there which could be cited here (Recognizing the reagent microbiome... by Parkhill et al. and Contamination in low-biomass microbiome studies: issues and recommendations by Weyrich et al 2018)

Line 144: The authors fail to describe experimental procedures where the use of positive controls are utilized including mock communities and instances where titrations of positive controls are used to determine the limit of detection of a given experiment (e.g. Kim, Dorothy, et al. "Optimizing methods and dodging pitfalls in microbiome research." *Microbiome* 5.1 (2017): 1-14. ; KatharoSeq enables high-throughput microbiome analysis from low-biomass samples; <https://doi.org/10.1093/femsec/fiz045>). This is an important strategy for also determining potential background kit contaminants vs cross contaminants.

Line 169: This is a clever name, but to an English language learner might be confusing at first. It might be reasonable to consider a name such as c-oligos...just a suggestion though

Line 242-251: There are a couple reviews on CoDa and specific methods for dealing with this (DOI: 10.1016/j.jannepidem.2016.03.003; <https://doi.org/10.3389/fmicb.2017.02224>) it might be useful to include some of these generalized approaches.

Line 252: Since the relative abundances of the 96 ISDs tested in the authors pipeline had considerable variation (Figure S2a), is it really possible to perform this normalization factor? It seems you would also have to account for the technical variation observed. If the authors could please clarify.

Line 297: Please include the actual sequences to these coligos in supplemental material to make it easy for other researchers to use this resource. I understand they're in the authors github but also include as a supplemental table or fasta.

Line 302. The more commonly used word here is ASV rather than ESV (dada2).

Line 308-311: Why is 97% similarity used instead of 100%? The previous sentence refers to ESVs which suggest 100% similarity. Is this not the case?

Line 328-337: Can the authors suggest sterilization or decontamination methods to ensure cleanliness of the low-cost reuse of materials

Line 340: Please list which kits are 384 well format. I am only familiar with kits which are compatible with 96 well plates. Many kits come with 4 96-well plates but this isn't a 384 well extraction per say.

Line 369-381: It would be great if authors could also describe some of the challenges with doing direct PCR. Studies which describe the total success rate across known DNA extraction and library

prep methods would be useful here. Direct PCR is in theory great but in practice generally has a much lower success rate.

Line 395: The (Marotz, Clarisse, et al. "DNA extraction for streamlined metagenomics of diverse environmental samples." *Biotechniques* 62.6 (2017): 290-293.) paper describes a comparison of robots used for DNA extraction in the context of microbiome sample processing. These higher throughput (96 samples) utilize magnetic bead based cleanups on the KingFisher robot.

Line 416: "DNA normalization" This section needs to be removed or significantly revised
□ Multiple studies have shown that final read counts can be predictive of initial starting microbial biomass (DOI: 10.1128/mSystems.00218-17 and <https://doi.org/10.3389/fmicb.2021.638231>). Normalization of DNA prior to PCR is not required and in fact, prohibits the ability to assess initial biomass estimates based on read counts. This strategy only works however if all samples are treated the same and no normalization occurs. Unless your microbial community is 100% bacterial/archaea, these normalization efforts are futile. The reason is that most microbiome samples will have some sort of eukaryote present. Since the euks won't amplify in the amplicon measure, any DNA quantification method will overestimate the amount of bacterial/archaeal DNA. If samples are host-associated such as from an animal or plant, this is even more confounded as these eukaryote cells will have orders of magnitude more DNA per cell as compared to bacterial cells.

Line 424: Even the best DNA quant methods have a relatively poor lower limit of detection. For instance, the Qubit HS kit can only detect in the pg level if using the full 20ul of DNA. This still equates to 100s-1000s of cell if pure microbial.

Line 443: This (<https://doi.org/10.2144/btn-2018-0192>) paper demonstrates that doing PCR reactions in singlet is better than triplicate. Please explain why duplicates are better considering the additional costs and lack of experimental evidence.

Line 602: its also very expensive on a per sample basis (Pipin prep)

Line 672-Line688: (PMID: 29482639) developed a protocol using PMA to reduce host cells in microbiome sampling ()

Line 737-747: Data needs to be uploaded to a commonly accessed database (ncbi, ENA, etc)

Recommendations: create a supplemental table with all coligos and any other

Supplemental SOP 3.1 steps 3-4 should be removed - they are not necessary and only confuse the analysis

□ See above comments

□ Additionally, since non-molecular biologists might be reading this, its important to point out and clarify some of the basic lab hygiene such as doing DNA extraction and PCR in separate rooms if possible - at the very least on different benches and especially different 'hoods'.

Steps 10.1-10.10

□ Did the researchers verify that these individual library cleanups are necessary for the analysis? Its

very likely that all samples could be pooled into a single final pool and then that pool processed through the bead cleanup. This would save substantially on cost and increase throughput. If they did not verify its required, I suggest adding a caveat replacement step to just indicate the alternative protocol.

Bioinformatics

Rather than 'ESVs', the general nomenclature is ASVs (amplified sequence variant). Others have referred to this also as sOTU (sub-operational taxonomic unit).

We appreciate the reviewers' comments and the chance to revise our manuscript in accordance with their concerns. We have addressed each concern below. Reviewer comments are in black, and our comments are in orange.

Reviewer comments:

Reviewer 1 (Comments for the Author):

The authors provide a discussion of the current state-of-the-art in establishing. In addition, new results are presented, focusing on the application of synthetic DNA standards for tracking of sample cross-contamination and "absolute" quantification of. The latter is an increasingly, albeit not yet sufficiently addressed, concern that relates to the compositionality of sequencing data, in the sense that they only provide information about ratios of taxon abundances (or relative abundances). This can be addressed by using the standards as anchors for read count normalization/transformation, such that comparative analyses between samples becomes valid.

Overall, the manuscript is very well written and provides an enjoyable and informative read. Although some parts cover relatively basic topics, I believe that it should be of interest to the reader. I do not have any major concerns/comments but would like to ask the authors to address my questions/comments below. All comments/questions can be considered as minor and should be easily addressable by the authors.

Thanks!

- With respect to the application of coligos for sample tracking and/or cross-contamination detection, it is stated that less than 0.01% of coligo reads were out of place (Line 192). It would be informative to also plot (probably as a Supplementary Figure) the distribution of the percentages of unexpected coligo reads per sample/well. Also, was any consistent contamination observed, this may point to maybe contamination due preparation of the coligo stocks?

Thanks for this suggestion. I added a supplemental table that displays the proportion of non-target reads for each well (Table S1). There is no obvious pattern of contamination, except perhaps that well H12 was very slightly contaminated in each replicate of our experiment. This could point to very minor cross-contamination during preparation of coligo stocks. I made a note to this effect in the caption of the table.

Further, it is stated that read counts varied across replicates (line 194-196). It is not clear to me how to interpret this; as all wells contain a single coligo (except for cross-contamination), the differences in read counts thus appear to be due to varying read depths per sample/well, rather than variable detection rates (e.g., due to differential PCR amplification) among oligos.

There are several potential causes for the variation in read count that we observed among coligos. First, minor pipetting differences could cause different amounts of template or reagents to be moved through the PCR process. Second, PCR could perform slightly differently for each coligo, for instance, due to variation in GC content. At the moment, I do not see a clear way to determine the relative influences of these two potential sources of variation without doing a larger experiment with more replicates.

In the Turlousse et al. 2018 paper (Line 197-200), differences between coligos were based on mixtures of three coligos each, such that the observed differences, ignoring compositionality, were attributed to differences in detection efficiencies between coligos in each of the mixtures. Please clarify this. This may also need to be clarified in Figure S2.

I revisited the Turlousse paper and they said, “Analysis of the abundances of each spike-in control across STMs (jh: sample tracking mix, which was the mixture of spike ins) further revealed a negative relationship between detection rate and amplicon G+C content (....) This indicated that GC content contributed to the perceived underestimation of specific sequences, consistent with its known effect on PCR efficiency”. We now mention this at line 205 within our paper, “Turlousse et al. (2018) reported variation in read count among their sample-provenance tracking oligos as well, and suggest that this variation is due to the necessary differences among oligos in nucleotide composition, including GC content, which have systematic effects on their abundances in sequence reads. This variation adds to mounting evidence that technically-derived variation is a significant source of noise in sequencing data. Indeed, in a recent study of the human gut microbiome, Ji et al. (2019) suggested that an abundance threshold exists below which technical variation drives among-sample differences in microbial abundances.” We added the last couple sentences of this passage to this new draft of the manuscript. We hope this sufficiently addresses this point.

Also, Figure S2 shows sum coligos with very low count, please briefly discuss; this is again due to differences in library normalization?

In this case, we suspect the low counts were due to pipetting error, because we see the same coligos quite often in our larger libraries. I added a note to this effect to the figure caption. In our more complex library, all coligos showed up with many thousands of counts, though there was a great deal of variation in read counts among coligos. This is at least partially due to differences in template quality and amount. For instance, in samples with little quality template DNA, coligos will take up much more sequencer bandwidth because they compose a greater proportion of the total amplicon pool. To sum up, we expect the causes of variation in read counts among coligos to be multifarious.

- The coligos developed by the authors for sample/cross-contamination tracking are very short. Please discuss its potential for preferential PCR amplification. Although this should not affect the performance for their intended purpose, that may be a question raised by researchers planning to adopt this system.

We added a sentence to line 220, where we suggest that preferential amplification did not seem to be an issue in our tests because coligos were always a very, very small proportion of the reads we observed. We also added a sentence to this passage stating, “If libraries include samples with very little template DNA, then coligo and internal standard DNA should be reduced so that the final library is not dominated by non-target amplicons.”

- At Line 186-187. Please describe the library preparation scheme (two-step/one-step, dual/single indexing). Although this is described in the Supplementary Methods, it helps to readers to get that information without needing to look at the Supplementary Methods.

Thanks, we added that detail.

- At Line 190. It is mentioned that reads are "mapped". Please briefly describe. Btw, it seems that exact matching was used for the short coligos, rather than "mapping".

I changed the word to “matched” to make this passage more clear.

- Also, it would be good to acknowledge that the level of cross-contamination that can be detected depends on the amount of coligos added to each sample/well, as touched upon in Tourlousse et al. 2018. This seems relevant to the discussion at Line 230-231.

This is a good point. I added the following at line 241 “The degree of cross-contamination that is detectable will depend upon the ratio of coligo DNA to template DNA. As more coligos are added, then more minor instances of cross-contamination will be detectable. We urge researchers that are interested in using coligos to consider the concentration that we used in our libraries as a starting point and modify that concentration as required. ”

- In line 202, the meaning of 10,000 "replicates" of samples is not clear. You mean 10,000 different samples. Also, please add the type of NovaSeq platform and read length, not critical but just for completeness, as well as the library preparation scheme (two-step/one-step, dual/single indexing).

Thanks for pointing this out. We reworded this passage to be more clear as, “... 10,000 replicates (including MID labeled PCR duplicates from over 5,000 samples)” and added the other details you suggested.

- In line 208, it seems that proportions (that is, total-sum-scaling to 1) are presented, rather than percentages used in other places (e.g., 0.01% as discussed in the previous comment). Please use to consistent total-sum-scaling basis, preferably as percentages as this may be easier to interpret.

We converted these proportions to percentages as you suggest

- In Line 306, it is stated that spurious ESVs may be due to degenerate primers. This is not clear to me. The primer sequences should be trimmed prior to analysis and thus not contribute the number of observed ESVs. Please clarify.

Thanks for mentioning this. Originally, we had not removed primers, thus leading to many ASVs. We now do this and have amended the passage accordingly.

- In Line 300-301, it is mentioned that the reads for the ISD showed quantitative behavior, shown as a linear relationship between ISD amount and read counts in Fig. S1. Please explain how the data counts per sample (I guess each different ISD concentration tested represents a different sample/library) were normalized, rarefaction?

No normalization was performed, rather raw read counts were observed. Note, that in practice, one would want to divide reads for each taxon by reads for the ISD (as we discuss at line 282). However, our goal here was to ensure that when we added more ISD to a sample then we would obtain more reads from that sample, and, moreover, that the shift in read count would follow expectations based on the change in ISD concentration in each sample.

Also, to my understanding, the relationship between added ISD amount added into a constant amount of mock DNA should display a hyperbolic dependency, from 0% of reads assigned to the ISD to 100% of reads assigned to the IDS. At lower amounts of ISDs added, this will appear as linear. On the other hand, expressing data as the ratio of ISD reads to mock reads should be linear, irrespectively of the range of ISD amounts tested. Basically,

$$\text{isdReads} = \text{isdCopies} / \text{totalCopies} * \text{totalReads} = (\text{isdCopies} / (\text{isdCopies} + \text{mockCopies})) * \text{totalReads}$$
 (hyperbole)

$$\text{isdReads} / \text{mockReads} = \text{isdCopies} / \text{mockCopies}$$
 (linear, assuming constant mock-Copies)

This is a comment; please consider.

Thank you for this comment. However, I am not sure that I understand. It seems that the equation for the hyperbola you provide there is no squared terms and the only way it differs from the linear equation is by a constant. Thus, both equations provide linear outcomes. However, perhaps I am misunderstanding your suggestion. Please correct me if I am not following.

- Line section 448-451, it is suggested that the number of PCR cycles should low, preferably less than 35. While this is true, the actual value very much depends on the polymerase and purity of the sample. Further, in my view, we should talk about reducing the amplicon yield (or fold amplification), rather than PCR cycles. Just a comment to consider.

This is an interesting consideration. My understanding of the argument for reducing PCR cycles is two fold. 1) more per cycles means more opportunity for polymerase to slip or otherwise introduce errors 2) as amplicons increase in abundance, low frequency amplicons (such as errors) could increase in frequency and become easier to detect. Reducing fold amplification should address my second concern, but not the first, necessarily. For instance, one could simply amplify less template to reduce total amplicon yield, but this comes with other concerns. Also reducing fold amplification would make it harder to detect rare, biologically-derived amplicons. I added the following sentence to this passage (line 497), “High cycle count can also lead to undesirable amplification of very rare template molecules, such as those derived from technical error. ”

- Although the use of long reads for analyzing microbiome by full-length rRNA operon sequencing is still in its infancy and may not provide the required throughput, a short paragraph on this topic may be helpful.

We agree that long-read tools are of great interest. However, our review is already really long, so we have decided to not add an additional paragraph. However, we did modify the following phrase in the introduction, slightly, to make it clear that we won't be discussing long-read tools, line 63 “Nor do we compare sequencing instruments (including new long-read machines), though the multiplexing advances we describe require the use of the latest generation of short-read sequencers (e.g., the Illumina NovaSeq).”.

- The others often employ NovaSeq for sequencing, because of its high throughput. However, as far as I know, such sequencing data may be more challenging to handle using state-of-the-art denoisers such as DADA2 as quality scores are binned. The authors should briefly discuss this.

Yes, we recently became aware that dada2 does not seem to handle NovaSeq data well. We added a mention of this to the supplementary bioinformatics. The new passage states, “We note that NovaSeq machines provide a binned quality score that is different from earlier Illumina machines. Thus dada2 (Callahan et al. 2016) and other denoisers, including USE-ARCH, are challenged by the new quality data. No clear consensus has yet emerged for how

to best deal with binned quality scores. In our view, the dramatic output provided by the NovaSeq makes it the sequencing machine of choice, despite the challenge posed by binned quality scores”

We also added the following to the introduction (line 69), “Notably, as new sequencing platforms are brought to market existing bioinformatics methods are challenged and can fail, thus researchers should expect to continually modify their bioinformatic pipeline.”. Regarding our own bioinformatics, we have moved forward while filtering reads as best we can using current methods.

- The authors suggest a read length of 2x250 bp in figure 1; however, not all NovaSeq platforms provide such read lengths. Also, even 2x250 bp reads may be too short for V3V4 amplicons, if read merging is part of the bioinformatics workflow. Please briefly discuss.

In accordance with your request, the following passage was added to Box 2, “We also note that the choice of primer and marker locus influences the read length desired from a sequencer. At the time of writing, NovaSeq machines can provide read lengths of up to 250 bases. Even when using paired reads, this may not be sufficient length to recover the whole marker locus from all organisms. In our work, we have noticed that this is a particular problem with using the ITS1 locus for fungi. We often cannot merge paired reads and must resort to concatenating reads or analyzing forward reads only.”

- In the Supplementary Material on Bioinformatics, it is stated that reads with more than a single error were removed. I guess this is based on the Expected Error that can be used as filtering criterion, please clarify and mention the term Expected Errors if appropriate as this is quite common.

The phrase “expected error” has been added to this passage.

Reviewer #2 (Comments for the Author):

"Characterizing microbiomes via sequencing of marker loci: techniques to improve throughput, account for cross-contamination, and reduce cost"

This is a review/opinion paper highlighting the challenges of consistency and data reproducibility across microbiome studies. This is an important subject and I think overall the paper does a good job reviewing the current literature on the subject. The paper also highlights the use of a new strategy for spike ins. There are a few parts of the paper which need revised to include additionally important research articles. One section which describes the need for DNA normalization prior to PCR needs to be either eliminated or significantly revised to address shortfalls of this approach. The authors have not made their data publicly available via the standard sequence databases (ncbi or ENA) and thus need to do so before acceptance. Lastly, the authors should consider making their novel oligos and overall primers even more clear and easy for other researchers to simply 'order' and use from IDT. At the current state, it's a little unclear how one would go about doing this as an outsider. I hope they can use this feedback to make the current pipeline more approachable for either new or seasoned microbiome researchers.

Thanks for the comments.

Main text:

Abstract Line 11-12: Interesting that they say nucleic acid extraction or library prep. It seems reasonable that including synthetic molecules at each step is better than one? Could the authors please elaborate? Most well-to-well contamination occurs during DNA extraction (Minich 2019).

We agree and amended the passage at line 252 to reflect this, “If desired, additional coligos could be designed and added to samples during library preparation, thus contamination during extraction could be distinguished from contamination during library preparation.”.

Intro Line 29-42: This is a great paragraph outlining the problems in the field. It would be fantastic if the authors provided a figure showing the results of this query. I understand they have written the detail in the text but a figure would be very appealing to a broad audience. In addition, it would be great if the authors included additional studies in this review dating back over the past 5 year (perhaps an additional 10-20 per year for 5 years) to see if there have been any trends of increased use of controls etc over that timeframe.

Thanks for the suggestion. We have added the figure that you suggest. and included a short survey looking back in time to see to what extent controls have become more commonly used. It seems that they haven't. In my own experience, I see controls more often in top journals, and have come to expect this. But a lot of papers still don't use negative controls of any sort, or at least don't document this in the methods.

Line 39: The newer sequencing instruments (Hiseq 4000, Novaseq, etc) use a patterned flow cell which does not work as well for low-diversity libraries. With Illumina taking the HiSeq 2500 offline, many researchers still continue to use the miseq as a standard instrument. It might be relevant to point this out as a challenge of doing any amplicon based approach on the Illumina platforms going forward.

By our understanding the problems inherent to low-diversity libraries are fairly similar across platforms. Here is a url from Illumina to this effect: <https://support.illumina.com/bulletins/2016/07/what-is-nucleotide-diversity-and-why-is-it-important.html> The salient portion of this web page is near the bottom and says that the amount of PhiX to add to a Novaseq and a Miseq run is about the same (5%), with maybe a bit more added to the Miseq (Illumina suggests 5–10% in this document, though we have seen other estimates elsewhere). Given that there doesn't seem to be a big difference among machines, at least from what we can gather, we decided to not add anything else to the manuscript. If you know of new work that suggests otherwise, then I would be very keen to see it! Currently, we make mention of the problems caused by low diversity libraries at line 561. Thanks for the comment.

Line 73: It would be great if the authors added a table outlining the various robots used in microbiome processing including DNA extraction and library prep along with prices. They do list some manufacturers but don't discuss this in more detail including MSRP list prices.

Thanks for the suggestion. Unfortunately, I think such a table would be outdated instantly. I started to make the table and it didn't feel like it would be very useful as it was not practical to be very thorough as there are a vast array of options available to consumers and price estimates are generally only available after contacting the company. These estimates vary depending on if the company is selling to industry or a University, and based on

estimated consumable sales, so providing price estimates seems potentially misleading and those estimates, in the best case, would be out of date in a matter of months. For these reasons, I decided not to include a table. Apologies!

Line 113-114: There are a few good reviews out there which could be cited here (Recognizing the reagent microbiome... by Parkhill et al. and Contamination in low-biomass microbiome studies: issues and recommendations by Weyrich et al 2018)

Thanks, we added these citations.

Line 144: The authors fail to describe experimental procedures where the use of positive controls are utilized including mock communities and instances where titrations of positive controls are used to determine the limit of detection of a given experiment (e.g. Kim, Dorothy, et al. "Optimizing methods and dodging pitfalls in microbiome research." *Microbiome* 5.1 (2017): 1-14. ; KatharoSeq enables high-throughput microbiome analysis from low-biomass samples; <https://doi.org/10.1093/femsec/fiz045>). This is an important strategy for also determining potential background kit contaminants vs cross contaminants.

Thank you for pointing out this oversight. The following passage has been added to the manuscript (line 421), "If possible, positive controls containing cells from taxa of known interest should be included during extraction. For exploratory work, a cellular mock community could be included as a positive control. Such mock communities could be made from available culture stocks or purchased (e.g., ZymoBIOMICS provides such an offering). A mock community can also confirm the results of bioinformatics (i.e., the number of ESVs obtained from bioinformatics matches or does not match expectations). Finally, positive controls can be titrated, such that the limit of detection for a particular number of cells or number of molecules can be approximately quantified (Minich et al. 2018)"

Line 169: This is a clever name, but to an English language learner might be confusing at first. It might be reasonable to consider a name such as c-oligos...just a suggestion though

Thanks, we decided to stick with coligos for this manuscript.

Line 242-251: There are a couple reviews on CoDa and specific methods for dealing with this (DOI: 10.1016/j.annepidem.2016.03.003; <https://doi.org/10.3389/fmichb.2017.02224>) it might be useful to include some of these generalized approaches.

Thanks. The latter of these citations was included in our manuscript at line 265, along with a few other papers describing various CoDa tools, e.g., Fernandes et al. 2014 and Tsilimigras and Fodor 2016

Line 252: Since the relative abundances of the 96 ISDs tested in the authors pipeline had considerable variation (Figure S2a), is it really possible to perform this normalization factor? It seems you would also have to account for the technical variation observed. If the authors could please clarify.

Thanks for asking for clarification here. We do not suggest that coligos should be used as ISDs, for the reason you mention. We added a note that explicitly states that at line, 204.

Line 297: Please include the actual sequences to these coligos in supplemental material to make it easy for other researchers to use this resource. I understand they're in the authors github but also include as a supplemental table or fasta.

We have now included the fasta that is on GitHub as a supplemental file.

Line 302. The more commonly used word here is ASV rather than ESV (dada2).

See our comment below. Thanks for your attention to detail.

Line 308-311: Why is 97% similarity used instead of 100%? The previous sentence refers to ESVs which suggest 100% similarity. Is this not the case?

Yes, we reworded this passage for clarity. The passage now reads, “To better determine ISD relative abundance, all ESVs that aligned to the ISD were summed for each replicate and this sum used in proportion calculations.”

Line 328-337: Can the authors suggest sterilization or decontamination methods to ensure cleanliness of the low-cost reuse of materials

We make mention of acid-washing beads for reuse (line 362). Unfortunately, however, we have found little information for how to reuse certain consumables. We did find some pipette tip washing tools, but little information showing how well they worked. I also found mixed results for autoclaving plastics; some people think nucleic acids are not always sufficiently broken down by autoclaving. In the absence of better science, I didn’t have much to report here. This seems like an important gap in knowledge. I did add a passage at line 350 that states, “Additionally, these steps typically require considerable expenditure of single-use, plastic consumables (i.e., pipette tips, micro-centrifuge tubes). We are aware of pipette tip washing tools (e.g., those made by Grenova, Richmond, VA), but these tools are currently unsuitable for pipette tips with filters. Moreover, many pipette tips, because of their filtering inserts, which are typically a different plastic than the body of the tip, are not recyclable, thus contributing to the large amount of laboratory waste generated worldwide (Urbina et al. 2015).”

Line 340: Please list which kits are 384 well format. I am only familiar with kits which are compatible with 96 well plates. Many kits come with 4 96-well plates but this isn’t a 384 well extraction per say.

I added one example parenthetically, the NucleoMag 384 kit from Takara. I have seen a few other options too and suspect these will become very common within a few years.

Line 369-381: It would be great if authors could also describe some of the challenges with doing direct PCR. Studies which describe the total success rate across known DNA extraction and library prep methods would be useful here. Direct PCR is in theory great but in practice generally has a much lower success rate.

Thanks for the suggestion. The papers that we cite by Flores and Videvall quantify the differences in performance between direct PCR and traditional extraction techniques. I spent a bit of time looking for papers criticizing direct PCR and, surprisingly, didn’t find much that was negative. Instead, everyone seems to be showing slightly different, but generally comparable results for direct PCR as compared to other methods. That said, the Kai paper we reference shows that for direct PCR to perform at its best some mechanical disruption of samples can help. If you can provide citations that demonstrate the challenges of direct PCR I would gladly include them here, but I haven’t found much, aside from the Kai paper.

Line 395: The (Marotz, Clarisse, et al. "DNA extraction for streamlined metagenomics of diverse environmental samples." *Biotechniques* 62.6 (2017): 290-293.) paper describes a

comparison of robots used for DNA extraction in the context of microbiome sample processing. These higher throughput (96 samples) utilize magnetic bead based cleanups on the KingFisher robot.

Thanks for this citation, we added it to the referenced passage.

Line 416: "DNA normalization" This section needs to be removed or significantly revised. Multiple studies have shown that final read counts can be predictive of initial starting microbial biomass (DOI: 10.1128/mSystems.00218-17 and <https://doi.org/10.3389/fmicb.2021.638231>). Normalization of DNA prior to PCR is not required and in fact, prohibits the ability to assess initial biomass estimates based on read counts. This strategy only works however if all samples are treated the same and no normalization occurs.

We agree that relative abundance data can be transformed into absolute abundance estimates (indeed, we state this in the second sentence of this passage [see below]), given that all samples are treated exactly the same and an ISD is used. The use of an ISD allows normalization of samples to a standard concentration (thus promoting more equitable distribution of sequencer bandwidth among samples), because the shift in ISD relative abundance can be used to back-transform relative abundances into absolute abundance estimates, see our previous paper in *Mol. Ecol. Resources* (The quest for absolute abundances...). We added a statement to this passage making this more clear (line 451) that states, "It is sometimes desirable to standardize the concentration of extracted DNA prior to PCR and sequencing. Otherwise samples with more DNA are expected to generate more amplicons and more sequence reads. We note that if DNAs are normalized to a standardized concentration then an ISD should be used if estimates of absolute abundances of microbes are desired (see above)." We hope this change addresses this point. In our experience, many (perhaps most) researchers perform some sort of normalization of samples prior to either PCR, or in some cases, after PCR and before sequencing. We agree with you that if absolute abundances are desired, normalization cannot be performed, except when an ISD is used. I should have made this more clear in the first draft, and I hope the subtle changes I made will help readers understand this critical point.

Unless your microbial community is 100% bacterial/archaea, these normalization efforts are futile. The reason is that most microbiome samples will have some sort of eukaryote present. Since the euks won't amplify in the amplicon measure, any DNA quantification method will overestimate the amount of bacterial/archaeal DNA. If samples are host-associated such as from an animal or plant, this is even more confounded as these eukaryote cells will have orders of magnitude more DNA per cell as compared to bacterial cells.

This is a good point. I added a sentence to this section stating this, which reads, "Normalization of samples that contain varying amounts of eukaryotic DNA can be challenging, as eukaryotic DNA often will not amplify during PCR."

Line 424: Even the best DNA quant methods have a relatively poor lower limit of detection. For instance, the Qubit HS kit can only detect in the pg level if using the full 20ul of DNA. This still equates to 100s-1000s of cell if pure microbial.

Agreed. Also taxa have different size genomes, so DNA quantification is not an appropriate method for the counting of cells. For normalizing libraries to more equitably assign sequencer bandwidth to samples, these methods do suffice. Stating this goal was our only

purpose in this passage. To make this more clear, I modified the last sentence of this passage so that it now reads, “To be clear, spectrophotometry and fluorometry measure total nucleic acids present, if measurements of amplicon concentration are desired then qPCR is a more appropriate approach (Nakayama et al 2016), and none of these tools can accurately estimate cell densities, due to a lack of resolution, CNV (Box 1), and differences in genome size among taxa.”

Line 443: This (<https://doi.org/10.2144/btn-2018-0192>) paper demonstrates that doing PCR reactions in singlet is better than triplicate. Please explain why duplicates are better considering the additional costs and lack of experimental evidence.

Thanks for this comment. I added the following statement to this passage, “Marotz et al. (2019) suggest that singlet PCR is sufficient; however, we have found duplicate PCR to be worth the additional cost because it provides more assurance that a sample will not be neglected due to minor errors during pipetting. Moreover, if PCR replicates are assigned unique MIDS, then performing PCR in duplicate can provide ample quantification of technical variation induced by library preparation.”

Line 602: its also very expensive on a per sample basis (Pipin prep)

True, but it is quite cheap if it can be done once at the end of a library preparation. I added a statement to this passage that makes it more clear that clean up should be done only once if possible. The statement reads, “ Unfortunately, library clean-up can be quite costly, so if the protocol allows, clean-up should be done only once on pooled, MID-labeled DNAs.”

Line 672-Line688: (PMID: 29482639)developed a protocol using PMA to reduce host cells in microbiome sampling ()

Cool, thanks! I added this citation and the following statement to this passage: “ A similar, but much more cost-effective, approach was demonstrated by Marotz et al (2018) that relies on selective lysis of mammalian cells followed by Propidium monoazide treatment (these authors estimate a \$0.15 USD cost per sample for this method).” Incidentally, if you have seen or heard of much work along these lines for plants, I would be keen to hear about it. In my own work, I typically recover an overwhelming amount of cpDNA from plants and have yet to find a solution that I really like.

Line 737-747: Data needs to be uploaded to a commonly accessed database (ncbi, ENA, etc)

Respectfully, I chose to upload data to a publicly accessed website hosted by our university instead of NCBI since the data are non-biological and I thought they would clutter up the standard databases. The biological data we analyzed that was from samples belonging to a variety of researchers has been uploaded to a publicly accessed website as well. Those data are partially the property of other researchers, who will likely place them in other databases. Thus, to avoid redundancy I decided to forgo accessioning the data in the NCBI SRA. The digital objects and websites that I used are institutional repositories and so should remain online and accessible for the foreseeable future. I can post the data to NCBI if desired, but, for the reasons mentioned above, I am not sure that doing so would be useful to readers or miners of the NCBI database.

Recommendations: create a supplemental table with all coligos and any other

Coligo sequences are now provided in a supplemental fasta, in addition to the GitHub repo.

Supplemental SOP 3.1 steps 3-4 should be removed - they are not necessary and only confuse the analysis see above comments. Additionally, since non-molecular biologists might be reading this, its important to point out and clarify some of the basic lab hygiene such as doing DNA extraction and PCR in separate rooms if possible - at the very least on different benches and especially different 'hoods'.

I combined steps 3-4 into step two and made them optional. For us, these steps are useful, because we are using an ISD. I hope the notes we added to the main text (see above) resolve your concerns. Also, I added a paragraph about good laboratory practice to the start of the SOP, as you suggest. Thanks.

Steps 10.1-10.10 Did the researchers verify that these individual library cleanups are necessary for the analysis? Its very likely that all samples could be pooled into a single final pool and then that pool processed through the bead cleanup. This would save substantially on cost and increase throughput. If they did not verify its required, I suggest adding a caveat replacement step to just indicate the alternative protocol.

Thanks. We are currently testing this. A caveat has been added. I also added a statement to the main text (line 642) stating that clean-up should be done once if possible. Additionally, thanks for your attention to detail and reading through our SOP! Much appreciated.

Bioinformatics Rather than 'ESVs', the general nomenclature is ASVs (amplified sequence variant). Others have referred to this also as sOTU (sub-operational taxonomic unit).

Still others have used "zero-radius OTUs (zOTUS)", and others have just stuck with OTU (including myself, in other papers). The Callahan paper arguing for the use ASVs/ESVs actually uses both terms in the same paper (exact sequence variant is in the title and elsewhere, but ASVs are used throughout). Unfortunately, I do not think there has been a consensus around which name to use. I personally would prefer to stick with OTU, and simply stipulate that each sequence variant is its own OTU, however this doesn't seem to be a viewpoint shared by many others! We have decided to mention that both ESV and ASV refer to the same thing (line 329) and retain our current use of ESV in much of the manuscript, since it is what we and our collaborators tend to use. If this is a sticking point, we can change this to ASV.

June 7, 2021

Dr. Joshua G Harrison
University of Wyoming
Laramie

Re: mSystems00294-21R1 (Characterizing microbiomes via sequencing of marker loci: techniques to improve throughput, account for cross-contamination, and reduce cost)

Dear Dr. Joshua G Harrison:

Thank you for your detailed resubmission and for choosing mSystems to publish this work. I anticipate that this piece will be valuable for both experienced and novice microbiome researchers.

Your manuscript has been accepted, and I am forwarding it to the ASM Journals Department for publication. For your reference, ASM Journals' address is given below. Before it can be scheduled for publication, your manuscript will be checked by the mSystems senior production editor, Ellie Ghatineh, to make sure that all elements meet the technical requirements for publication. She will contact you if anything needs to be revised before copyediting and production can begin. Otherwise, you will be notified when your proofs are ready to be viewed.

We recognize that the video files can become quite large, and so to avoid quality loss ASM suggests sending the video file via <https://www.wetransfer.com/>. When you have a final version of

the video and the still ready to share, please send it to Ellie Ghatineh at eghatineh@asmusa.org.

Sincerely,

Peter Turnbaugh
Editor, mSystems

Journals Department
Supplemental Figure 3: Accept
Supplemental Methods: Accept
Supplemental Figure 2: Accept
Supplemental Figure 1: Accept
Supplemental Table 1: Accept
Supplemental Figure 4: Accept
Supplemental Material: Accept